# Private Multi-Task Learning: Formulation and Applications to Federated Learning

**Shengyuan Hu**                                                                                     *shengyuanhu@cmu.edu*
*Carnegie Mellon University*
**Zhiwei Steven Wu**                                                                                 *zstevenwu@cmu.edu*
*Carnegie Mellon University*
**Virginia Smith**                                                                                   *smithv@cmu.edu*
*Carnegie Mellon University*

**Reviewed on OpenReview:** *https: // openreview. net/ forum? id= onufdyHvqN*

## Abstract

Many problems in machine learning rely on *multi-task learning (MTL)*, in which the goal is to solve multiple related machine learning tasks simultaneously. MTL is particularly relevant for privacy-sensitive applications in areas such as healthcare, finance, and IoT computing, where sensitive data from multiple, varied sources are shared for the purpose of learning. In this work, we formalize notions of client-level privacy for MTL via *billboard privacy* (BP), a relaxation of differential privacy for mechanism design and distributed optimization. We then propose an algorithm for mean-regularized MTL, an objective commonly used for applications in personalized federated learning, subject to BP. We analyze our objective and solver, providing certifiable guarantees on both privacy and utility. Empirically, we find that our method provides improved privacy/utility trade-offs relative to global baselines across common federated learning benchmarks.

## 1 Introduction

Multi-task learning (MTL) aims to solve multiple learning tasks simultaneously while exploiting similarities/differences across tasks (Caruana, 1997). MTL is commonly used in applications that warrant strong privacy guarantees. For example, MTL has been used in healthcare, as a way to learn over diverse populations or between multiple institutions (Baytas et al., 2016; Suresh et al., 2018; Harutyunyan et al., 2019); in financial forecasting, to combine knowledge from multiple indicators or across organizations (Ghosn & Bengio, 1997; Cheng et al., 2020); and in IoT computing, as an approach for personalized federated learning (Smith et al., 2017; Hanzely & Richtárik, 2020; Hanzely et al., 2020; Ghosh et al., 2020; Sattler et al., 2020; Deng et al., 2020; Mansour et al., 2020). While MTL can significantly improve accuracy when learning in these applications, there is a dearth of work studying the privacy implications of multi-task learning.

In this work, we develop and theoretically analyze methods for MTL with formal privacy guarantees. Motivated by applications in federated learning, we aim to provide **client-level privacy**, where each task corresponds to a client/user/data silo, and the goal is to protect sensitive information in each task's data (McMahan et al., 2018). We focus on ensuring *differential privacy* (DP) (Dwork et al., 2006), which (informally) requires an algorithm's output to be insensitive to changes in any single entity's data.

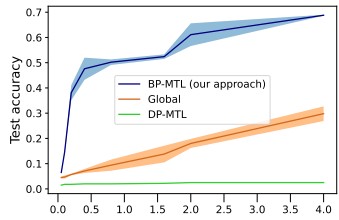

Figure 1: Naively using current client-level DP formulations with MTL results in models that are no better than a random guess.

For MTL, where a separate model is generated for each client, using client-level DP directly would require the entire set of predictive models across all tasks to be insensitive to changes in the private data of any single task. This

requirement is too stringent for most applications, as it implies that the predictive model for task $k$ must have little dependence on the training data for task $k$, thus preventing the usefulness of the model (see Figures 1).

To address this issue, we leverage a privacy model known as the *billboard model* (Hsu et al., 2016b). The *billboard model* is built using: (1) a global signal from a differentially private process that is public to all the clients, and (2) every client $i$'s private data. Unlike DP, the *billboard model* ensures that for each task $k$, the set of output predictive models for all other tasks *except* $k$ is insensitive to $k$'s private data.[1] Therefore, it allows the predictive model for task $k$ to depend on $k$'s private data, helping to preserve each task's utility.

In this work, we develop new learning algorithms for MTL that satisfy the billboard model with rigorous privacy and utility guarantees. Specifically, we propose Private Mean-Regularized MTL, a simple framework for learning multiple tasks while ensuring client-level privacy. We show that our method achieves $(\epsilon, \delta)$-*billboard privacy (BP)* (Defined in Section 3.2). Our scalable solver builds on FedAvg (McMahan et al., 2017), a common method for communication-efficient federated optimization. We analyze the convergence of our solver on both nonconvex and convex objectives, demonstrating a tradeoff between privacy and utility, and evaluate this trade-off empirically on multiple federated learning benchmarks. We summarize our contributions below:

- We propose Private Mean-Regularized MTL, a simple MTL framework that provides **client-level billboard privacy (BP)** (Section 4). We prove that our method achieves $(\epsilon, \delta)$-BP.

- We analyze the convergence of our communication-efficient solver on convex and nonconvex objectives. Our convergence analysis extends to non-private settings with partial participation, which may be of independent interest for problems in cross-device federated learning.

- Finally, we explore the performance of our approach on common federated learning benchmarks (Section 5). Our results show that we can retain the accuracy benefits of MTL in these settings relative to global baselines while still providing meaningful privacy guarantees. Further, even in cases where the MTL and global objectives achieve similar accuracy, we find that privacy/utility benefits exist when employing our private MTL formulation compared to privately learning a single global model.

## 2 Background and Related Work

**Multi-task learning.** Multi-task learning considers jointly solving multiple related ML tasks. Our work focuses on the general and widely-used formulation of multi-task relationship learning (Zhang & Yeung, 2010), as detailed in Section 3. This form of MTL is particularly useful in privacy-sensitive applications where datasets are split among multiple heterogeneous entities (Baytas et al., 2016; Smith et al., 2017; Ghosn & Bengio, 1997). In these cases, it is natural to view each data source (e.g., financial institution, hospital, mobile phone) as a separate 'task' that is learned in unison with the other tasks. This allows learning to be performed jointly, but the models to be personalized to each data silo. For example, in the setting of cross-device federated learning, MTL is commonly used to train a personalized model for each device in a distributed network (Smith et al., 2017; Liu et al., 2017).

**Federated learning.** A motivation for our work is the application of federated learning (FL), in which the goal is to collaboratively learn from a number of private data silos, such as remote devices or servers (McMahan et al., 2017; Kairouz et al., 2019; Li et al., 2020a). To ensure client-level DP in FL, a common technique is to learn one *global model* across the distributed data and then add noise to the aggregated model to sufficiently mask any specific client's update (e.g., Kairouz et al., 2019; McMahan et al., 2018; Geyer et al., 2017; Levy et al., 2021; Lowy & Razaviyayn, 2021; Lowy et al., 2022). However, a defining characteristic of federated learning is that the distributed data are likely to be heterogeneous, i.e., each client may generate data via a distinct data distribution (Kairouz et al., 2019; Li et al., 2020a). To model the (possibly) varying data distributions on each client, it is natural to instead consider learning a separate model for each client's local dataset. To this end, a number of recent works have explored multi-task learning as a way to improve the accuracy of learning in federated networks (Smith et al., 2017; Hanzely & Richtárik, 2020; Hanzely et al., 2020; Ghosh et al., 2020; Sattler et al., 2020; Deng et al., 2020; Mansour et al., 2020). Despite the

---

[1]This privacy guarantee is known as joint differential privacy (JDP) Kearns et al. (2014), and billboard privacy is a common way to achieve JDP.

prevalence of multi-task federated learning, we are unaware of any work that has explored client-level privacy for commonly-used multi-task relationship models (Section 3) in federated settings.

**Differentially private MTL.** Prior work in private MTL differs from our own either in terms of the privacy formulation or MTL objective. For example, Wu et al. (2020) explore a specific MTL setting where a shared private feature representation is first learned, followed by task-specific models. We instead study multi-task relationship learning (Section 3), which is a general and widely-used MTL framework, particularly in federated learning (Smith et al., 2017). While our work focuses on client-level privacy, there has been work on data-level privacy for MTL, which aims to protect any single sample of local data rather than protecting the entire local dataset. For example, Xie et al. (2017) propose a method for data-level privacy by modeling each task as a sum of a public shared weight and a task-specific weight that is only updated locally, and Gupta et al. (2016) study data-level privacy for a mean estimation MTL problem. Li et al. (2019) study multiple notions of DP for meta-learning. Although similarly motivated by personalization, their framework does not cover the multi-task setting, where there exists a separate model for each task. Hu et al. (2020) studied example-level private multi-task learning but only their method is restricted to small scale convex task, which is different from our focus on client-level privacy. More closely related to our work, Jain et al. (2021) study a personalization method that learns a private shared representation. Although they similarly leverage the billboard model, their formulation cannot be applied to the general form of multi-task learning in this work. Their results are also limited to the special case of linear regression, unlike the broad set of convex and nonconvex objectives considered herein. Finally, Bietti et al. (2022) similarly propose a personalized federated learning method using the billboard model. Although both methods train a global model shared across tasks, the concrete algorithms are rather different. The main algorithm of Bietti et al. (2022) builds on Federated Residual Learning (Agarwal et al., 2020) while our main algorithm builds on mean-regularized multi-task learning. We provide an in-depth discussion and empirical comparison to this method in Appendix A.8.

## 3 Multi-Task Learning Setup and Privacy Formulation

In this section, we first formalize the multi-task learning objectives of interest (Section 3.1), and then discuss our proposed privacy formulation (Section 3.2).

### 3.1 Problem Setup

Multi-task learning aims to improve generalization by jointly solving and exploiting relationships between multiple tasks Caruana (1997); Ando & Zhang (2005). The classical setting of multi-task relationship learning (Zhang & Yang, 2017; Zhang & Yeung, 2010) considers $m$ different tasks with their own task-specific data, learned jointly through the following objective:

$$\min_{W,\Omega} \left\{ F(W,\Omega) = \left\{ \frac{1}{m} \sum_{k=1}^{m} \sum_{i=1}^{n_k} l_k(x_i, w_k) + \mathcal{R}(W,\Omega) \right\} \right\}. \tag{1}$$

Here $w_k$ is model for task $k$, $\{x_1, \ldots, x_{n_k}\}$ is the local data for the $k^{th}$ task, $l_k(\cdot)$ is the empirical loss, $W = [w_1; \cdots; w_m]$, and $\Omega \in \mathbb{R}^{m \times m}$ characterizes the relationship between every pair of tasks. A common choice for setting the regularization term $\mathcal{R}(W, \Omega)$ in prior works (Zhang & Yeung, 2010; Smith et al., 2017) is:

$$\mathcal{R}(W,\Omega) = \lambda_1 \mathrm{tr}(W\Omega W^T),$$

where $\Omega$ can be viewed as a covariance matrix, used to learn/encode positive, negative, or unrelated task relationships. In this paper, we focus primarily on the mean-regularized multi-task learning objective (Evgeniou & Pontil, 2004): a special case of (1) where $\Omega = (\mathbf{I_{m \times m}} - \frac{1}{\mathbf{m}} \mathbf{1_m} \mathbf{1_m^T})^2$ is fixed. Here $\mathbf{I_{m \times m}}$ is the identity matrix of size $m \times m$ and $\mathbf{1_m} \in \mathbb{R}^{\mathbf{m}}$ is the vector with all entries equal to 1. By picking $\lambda_1 = \frac{\lambda}{2}$, we can rewrite Objective 1 as:

$$\min_{W} \left\{ F(W) = \left\{ \frac{1}{m} \sum_{k=1}^{m} \left( \frac{\lambda}{2} \|w_k - \bar{w}\|^2 + \sum_{i=1}^{n_k} l_k(x_i, w_k) \right) \right\} \right\}, \tag{2}$$

where $\bar{w}$ is the average of task-specific models, i.e., $\bar{w} = \frac{1}{m} \sum_{i=1}^{m} w_k$. Note that $\bar{w}$ is shared across all tasks, and each $w_k$ is kept locally for task learner $k$. During optimization, each task learner $k$ solves:

$$\min_{w_k} \left\{ f_k(w_k; \bar{w}) = \frac{\lambda}{2} \|w_k - \bar{w}\|^2 + \sum_{i=1}^{n_k} l_k(x_i, w_k) \right\}. \tag{3}$$

**Application to Federated Learning.** In federated learning, where the goal is to learn over a set of $m$ clients in a privacy-preserving manner, initial approaches focused on learning a single global model across the data McMahan et al. (2017). However, as data distributions may differ from one client to another, MTL has become a popular alternative that enables every client to collaborate and learn a **separate, personalized** model of its own Smith et al. (2017). Specifically, in the case of mean-regularized MTL, each client solves Equation 3 and utilizes $w_k$ as its final personalized model. Unlike finetuning from a global model, MTL itself learns a separate model for each client by solving Objective 1 in order to improve the generalization performance (Zhang & Yang, 2017; Zhang & Yeung, 2010), which is not equivalent to simple finetuning from a global model (see Section 5). Despite the prevalence of mean-regularized multi-task learning and its recent use in applications such as federated learning with strong privacy motivations (e.g., Smith et al., 2017; Hanzely & Richtárik, 2020; Hanzely et al., 2020; Dinh et al., 2020), we are unaware of prior work that has formalized client-level differential privacy in the context of solving Objective 2.

### 3.2 Privacy Formulation

To consider privacy for MTL, we start by introducing the definition of *differential privacy (DP)* and then discuss its generalization to *joint differential privacy (JDP)*. In the context of multi-task learning, each of the $m$ task learners owns a private dataset $D_i \in \mathcal{U}_i \subset \mathcal{U}$. We define $D = \{D_1, \cdots, D_m\}$ and $D' = \{D'_1, \cdots, D'_m\}$, and call two sets $D, D'$ *neighboring sets* if they only differ on the index $i$, i.e., $D_j = D'_j$ for all $j$ except $i$.

**Definition 1** (Differential Privacy (DP) for MTL (Dwork et al., 2006)). *A randomized algorithm $\mathcal{M} : \mathcal{U}^m \to \mathcal{R}^m$ is $(\epsilon, \delta)$-differentially private if for every pair of neighboring sets that only differ in arbitrary index $i$: $D, D' \in \mathcal{U}$ and for every set of subsets of outputs $S \subset \mathcal{R}$,*

$$Pr(\mathcal{M}(D) \in S) \le e^\epsilon Pr(\mathcal{M}(D') \in S) + \delta. \tag{4}$$

In the context of MTL, an algorithm outputs one model for every task. In this work we are interested in studying *client-level differential privacy*, where the purpose is to protect one task's data from leakage to any other task McMahan et al. (2017). As mentioned previously and illustrated in Figure 2, since the output of MTL is a *collection of models*, traditional client-level DP would require that all the models produced by an MTL algorithm are insensitive to changes that happen in the private dataset of *any* single client/task.

**Why can't we apply traditional client-level DP?** With the above definition in mind, note that DP has a severe restriction: the model of any task learner must also be insensitive to changes in *its own data*, effectively rendering each model useless. Although it is intuitive that this would result in unacceptable performance, we verify it empirically in Figure 1. For a common federated learning benchmark (FEMNIST, discussed in Section 5), we apply DPSGD on the joint model that concatenates all clients' parameters. We compare MTL with vanilla client-level DP relative to training a global model with client-level DP and our proposed MTL approach using BP (below). With the naive DP formulation, MTL is significantly worse than the other approaches—improving only marginally upon random guessing.

**Billboard Privacy.** To overcome this limitation of traditional DP, motivated by the billboard model (Hsu et al., 2016b), we propose *billboard privacy (BP)*, a relaxed notion of DP, to formalize the client-level privacy guarantees for MTL algorithm. We provide the formal definition below.

**Definition 2** (Billboard Privacy (BP) (Hsu et al., 2016b)). *Consider any set of functions: $f_i : \mathcal{U}_i \times \mathcal{R} \to \mathcal{R}'$ and $g : \mathcal{U} \to \mathcal{R}$, a randomized algorithm $\mathcal{M} : \mathcal{U}^m \to \mathcal{R}^m$ represented as $[f_i(\Pi_i D, g(D))]^m$ is $(\epsilon, \delta)$-billboard private if for every $i$, for every pair of neighboring datasets that only differ in index $i$: $D, D' \in \mathcal{U}^m$ and for every set of subsets of outputs $S \subset \mathcal{R}^m$,*

$$Pr(\mathcal{M}(D)_{-i} \in S) \le e^\epsilon Pr(\mathcal{M}(D')_{-i} \in S) + \delta, \tag{5}$$

*where $\Pi_i D$ is $D$'s projection onto the $i$-th index and $\mathcal{M}(D)_{-i}$ represents the vector $\mathcal{M}(D)$ with the $i$-th entry removed.*

BP allows the predictive model for task $k$ to depend on the private data of $k$, while still providing a strong guarantee. BP provides $m-1$-out-of-$m$ privacy under Shamir's scheme of secret sharing (Shamir, 1979): even if all the other $m-1$ collude and share their information, they still will not be able to learn much about the private data in the task $k$. BP has mostly been used in applications related to mechanism design (Hsu et al., 2016a; Kannan et al., 2015; Hsu et al., 2016b). Although it is a natural choice for achieving client-level privacy in MTL, we are unaware of any work that studies the general MTL formulations considered herein subject to BP. We also note that we can naturally connect billboard privacy to standard differential privacy. Informally, if $g$ is $(\epsilon, \delta)$-differentially private, then $[f_i(\Pi_i D, g(D))]^m$ is $(\epsilon, \delta)$-billboard private for arbitrary $\{f_i\}_{i \in [1:m]}$ by definition of billboard privacy. In other words, if we take the output of a differentially private process and run some algorithm on top of that locally for each task learner *without* communicating to the global learner or other task learners, this whole process can be shown to be BP.

**Remark (Generality of Privacy Formulation).** Finally, note that our privacy formulation itself is not limited to the multi-task relationship learning framework. For any form of multi-task learning where each task-specific model is obtained by training a combination of global component and local component(e.g. Li et al. (2021)), we can provide a BP guarantee for the MTL training process by using a differentially private global component.

**Connection to Joint Differential Privacy.** Compared to BP, a weaker yet more general privacy formulation is known as *joint differential privacy (JDP)* (Kearns et al., 2014) defined formally below. By definition, $(\epsilon, \delta)$-BP implies $(\epsilon, \delta)$-JDP. Different from billboard privacy where a global differentially private message $g(D)$ is needed, JDP does not need any global information shared across all the clients. Compared to JDP, achieving BP is a harder problem since it requires learning a shared message (in the case of MTL, a private global model) that could be used for all $m$ clients while a JDP mechanism does not necessarily produce such message.

**Definition 3** (Joint Differential Privacy (JDP) (Kearns et al., 2014))**.** *A randomized algorithm $\mathcal{M} : \mathcal{U}^m \to \mathcal{R}^m$ is $(\epsilon, \delta)$-joint differentially private if for every $i$, for every pair of neighboring datasets that only differ in index $i$: $D, D' \in \mathcal{U}^m$ and for every set of subsets of outputs $S \subset \mathcal{R}^m$,*

$$Pr(\mathcal{M}(D)_{-i} \in S) \leq e^\epsilon Pr(\mathcal{M}(D')_{-i} \in S) + \delta, \tag{6}$$

*where $\mathcal{M}(D)_{-i}$ represents the vector $\mathcal{M}(D)$ with the $i$-th entry removed.*

## 4 PMTL: Private Multi-Task Learning

We now present PMTL, a method for joint differentially-private MTL (Section 4.1). We provide both a formal privacy guarantee (Section 4.2) and utility guarantee (Section 4.3) for our approach.

### 4.1 Algorithm

We summarize our solver for private multi-task learning in Algorithm 1. Our method is based off of FedAvg (McMahan et al., 2017), a communication-efficient method widely used in federated learning. FedAvg alternates between two steps: (i) each task learner selected at one communication round solves its own local objective by running stochastic gradient descent for $E$ iterations and sending the updated model to the global learner; (ii) the global learner aggregates the local updates and broadcasts the aggregated mean. By performing local updating in this manner, FedAvg has been shown to empirically reduce the total number of communication rounds needed for convergence in federated settings relative to baselines such as mini-batch FedSGD (McMahan et al., 2017). Our private MTL algorithm differs from FedAvg in that: (i) instead of learning a single global model, all task learners collaboratively learn separate, personalized models for each task; (ii) each task learner solves the local objective with the mean-regularization term; (iii) individual model updates are clipped and random Gaussian noise is added to the aggregated model updates to ensure client-level privacy.

In this work, we focus on providing global client-level billboard privacy. Therefore, we assume that we have access to a trusted global learner while aggregating updates from each task, i.e., it is safe for some global entity to observe/collect the individual model updates from each task. This is a standard assumption in federated learning, where access to a trusted server is assumed in order to collect client updates (Kairouz et al., 2019). To add an additional layer on security, our method has a natural extension to support secure

---

**Algorithm 1** PMTL: Private Mean-Regularized MTL

---

1: **Input:** $m$, $T$, $\lambda$, $\eta$, $\{w_1^0, \cdots, w_m^0\}$, $\widetilde{w}^0 = \frac{1}{m} \sum_{k=1}^m w_k^0$
2: **for** $t = 0, \cdots, T-1$ **do**
3:     Global Learner randomly selects a set of tasks $S_t$ and broadcasts the mean weight $\widetilde{w}^t$
4:     **for** $k \in S_t$ in parallel **do**
5:         Each client updates its weight $w_k$ for $E$ iterations, $o_k$ is the last iteration task $k$ is selected

6:         Each client sends $g_k^{t+1} = w_k^{t+1} - w_k^t$ back to the global learner.
7:     **end for**
8:     Global Learner computes a noisy aggregator of the weights

$$\widetilde{w}^{t+1} = \widetilde{w}^t + \frac{1}{|S_t|} \sum_{k \in S_t} g_k^{t+1} \min\left(1, \frac{\gamma}{\|g_k^{t+1}\|_2}\right) + \mathcal{N}(0, \sigma^2 \mathbf{I_{d \times d}})$$

9: **end for**
10: **Output** $w_1, \cdots, w_m$ as differentially private personalized models

---

11: ClientUpdate(w)
12:     **for** $j = 0, \cdots, E-1$ **do**
13:         Task learner performs SGD locally

$$w = w - \eta(\nabla_w l_k(w) + \lambda(w - \widetilde{w}^t))$$

14:     **end for**

---

aggregation, a common cryptographic primitive used in federated learning (Bonawitz et al., 2016; 2019; Kairouz et al., 2021). While SA/MPC are important to ensure secrecy while communicating model updates, neither trusted server assumption nor secure aggregation protects a client's private data from leakage to other clients by observing the model output. Thus, our algorithm focuses on addressing privacy concern for model personalization in federated learning.

There are several ways to overcome this privacy risk and thus achieve $(\epsilon, \delta)$-differential privacy. In this paper, we use the Gaussian Mechanism (Dwork & Roth, 2014) during global aggregation as a simple yet effective method, highlighted in **red** in line 8 of Algorithm 1. In this case, each client receives a noisy aggregated global model, making it difficult for any task to leak private information to the others. To apply the Gaussian mechanism, we need to bound the $\ell_2$-sensitivity of each local model update that is communicated to lie in $\mathcal{B} = \{\Delta w \| \|\Delta w\|_2 \leq \gamma\}$, as highlighted in **blue** in line 8 of Algorithm 1. Hence, at each communication round, the global learner receives the model updates from each clients, and clips the model updates to $\mathcal{B}$ before aggregation. Note that different from DPSGD (Abadi et al., 2016), when we solve the local objective for each selected task at each communication round, our algorithm doesn't clip and perturb the gradient used to update the task-specific model. Instead, since the purpose is to protect task or client-level privacy in multi-task learning, we perform standard SGD locally for each task and only clip and perturb the model update that is sent to the global learner. We formalize the privacy guarantee of Algorithm 1 in Section 4.2.

## 4.2 Privacy Analysis

We now rigorously explore the privacy guarantee provided by Algorithm 1. In our optimization scheme, for each task $k$, at the end of each communication round, a shared global model is received. After that the task specific model is updated by optimizing the local objective. We formalize this local task learning process as $h_k : \mathcal{D}_k \times \mathcal{W} \to \mathcal{W}$. Here we simply assume $\mathcal{W} \subset \mathbb{R}^d$ is closed. Define the mechanism for communication round $t$ to be

$$\mathcal{M}^t(\{D_i\}, \{h_i(\cdot)\}, \widetilde{w}^t, \sigma) = \widetilde{w}^t + \frac{1}{|S_t|} \sum_{k \in S_t} h_k(D_k, \widetilde{w}^t) + \beta^t, \tag{7}$$

where $\beta^t \sim \mathcal{N}(0, \sigma^2 \mathbf{I}_{d \times d})$. Note that $\mathcal{M}^t$ characterizes a Sampled Gaussian Mechanism given $\widetilde{w}^t$ as a fixed model rather than the output of a composition of $\mathcal{M}^j$ for $j < t$. To analyze the privacy guarantee

of Algorithm 1 over $T$ communication rounds, we define the composition of $\mathcal{M}^1$ to $\mathcal{M}^T$ recursively as $\mathcal{M}^{1:T} = \mathcal{M}^T(\{D_i\}, \{h_i(\cdot)\}, \mathcal{M}^{T-1}, \sigma)$.

**Theorem 1.** *Assume $|S_t| = q$ for all $t$ and the total number of communication rounds is $T$. There exists constants $c_1, c_2$ such that for any $\epsilon < c_1 \frac{q^2}{m^2} T$, the mechanism $\mathcal{M}^{1:T}$ is $(\epsilon, \delta)$ client-level differentially private for any $\delta > 0$ if we choose $\sigma \geq c_2 \frac{\gamma \sqrt{T \log(1/\delta)}}{\epsilon m}$. When $q = m$, $\mathcal{M}^{1:T}$ is $(\epsilon, \delta)$ client-level differentially private if we choose $\sigma = \frac{4\gamma \sqrt{T \log(1/\delta)}}{\epsilon m}$.*

Theorem 1 provides a provable privacy guarantee on the learned model. When all tasks participate in every communication round, i.e. $q = m$, the global aggregation step in Algorithm 1 reduce to applying Gaussian Mechanism without sampling rather than Sampled Gaussian Mechanism on the average model updates. We provide a detailed proof of Theorem 1 in Appendix A.1. Note in particular that Theorem 1 doesn't rely on how task learners optimize their local objective. Hence, Theorem 1 is not limited to Algorithm 1 and could be generalized to other local objectives and other global aggregation methods that produce a single model aggregate.

Now we show that Algorithm 1, which outputs $m$ separate models, satisfies BP. Given $\widetilde{w}^t$ for any $t \leq T$, we formally define the process that each task learner $k$ optimize its local objective to be $h'_k : \mathcal{D}_k \times \mathcal{W} \to \mathcal{W}$. Note that $h'_k$ is not restricted to be $h_k$ and could represent the optimization process for any local objective. In our MR-MTL problem, the average model broadcast by the global learner at every communication round is the output of a differentially private learning process. Task learners then individually train their models on the respective private data to obtain personalized models. By definition of billboard privacy in Section 3.2, we are able to show that Algorithm 1 satisfies BP:

**Corollary 2.** *There exists constants $c_1, c_2$, for any $0 < \epsilon < c_1 \frac{q^2}{m^2} T$ and $\delta > 0$, let $\sigma \geq \frac{c_2 \gamma \sqrt{T \log(1/\delta)}}{\epsilon m}$. Algorithm 1 that outputs $h'_k(D_k, \mathcal{M}^{1:T})$ for each task is $(\epsilon, \delta)$-billboard private.*

From Theorem 2, for any fixed $\delta$, the more tasks involved in the learning process, the smaller $\sigma$ we need in order to keep the privacy parameter $\epsilon$ the same. In other words, less noise is required to keep the task-specific data private. When we have infinitely many tasks ($m \to \infty$), we have $\sigma \to 0$, in which case only a negligible amount of noise is needed for the model aggregates to make the global model private to all tasks. We provide a detailed proof in Appendix A.1.

**Remark (Generality of Corollary 2).** Note that the privacy guarantee provided by Corollary 2 is not limited to mean-regularized MTL. For any form of multi-task relationship learning with fixed relationship matrix $\Omega$, as long as we fix the $\ell_2$-sensitivity of model updates and the noise scale of the Gaussian mechanism applied to the statistics broadcast to all task learners, the privacy guarantee induced by this aggregation step is fixed, regardless of the local objective being optimized. For example, as a natural extension of mean-regularized MTL, consider the case where task learners are partitioned into fixed clusters and optimize the mean-regularized MTL objective within each cluster, as in Evgeniou et al. (2005). In this scenario, Theorem 2 directly applies to the algorithm run on each cluster.

### 4.3 Convergence Analysis

As discussed in Section 3, we are interested in the following task-specific objective:

$$f_k(w_k; \widetilde{w}) = l_k(w_k) + \frac{\lambda}{2} \|w_k - \widetilde{w}\|_2^2 \tag{8}$$

where $\widetilde{w}$ is an estimate for the average model $\overline{w}$; $l_k(w_k) = \sum_{i=1}^{n_k} l_k(x_i, w_k)$ is the empirical loss for task $k$; $w_k \in \mathbb{R}^d$.

Here, we analyze the convergence behavior in the setting where a set $S_t$ of $q$ tasks participate in the optimization process at every communication round. Further, we assume the total number of communication round $T$ is divisible by the number of local optimization steps $E$: $T = 0 \mod E$. We present the following convergence result:

**Theorem 3** (Convergence under nonconvex loss (Informal)). *Let $f_k$ be $(L + \lambda)$-smooth. Assume $f_k$ is $G$-Lipschitz in $\ell_2$ norm such that $\gamma \geq G$. Further let $f_k^* = \min_{w, \bar{w}} f_k(w; \bar{w})$, $p = \frac{q}{m}$, and $B =$*

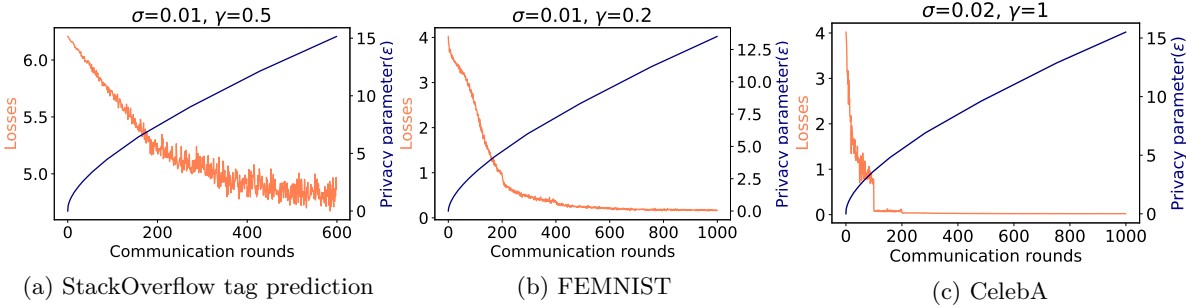

Figure 2: Loss and privacy parameter vs. communication rounds for PMTL. The blue line shows the change of $\epsilon$ in terms of number of communication rounds during training. The orange line shows the average training loss.

$\max_t \max_k f_k(w_k^t; \widetilde{w}^t)$. *If we use a fixed learning rate* $\eta_t = \eta = \frac{1}{pL + (p + \frac{1}{p})\lambda}$, *Algorithm 1 satisfies:*

$$\frac{1}{mT}\sum_{t=0}^{T-1}\sum_{k=1}^{m}\|\nabla f_k(w_k^t; \widetilde{w}^t)\|^2 \le \mathcal{O}\left(\frac{\lambda}{T}\right) + \mathcal{O}\left(\frac{\lambda B}{E}\right) + \mathcal{O}\left(\frac{d\lambda\sigma^2}{E}\right). \tag{9}$$

*Let* $\sigma$ *chosen as we set in Theorem 2. Take* $T = \mathcal{O}\left(\frac{m}{\lambda d\gamma^2}\right)$, *the right hand side is bounded by*

$$\frac{1}{mT}\sum_{t=0}^{T-1}\sum_{k=1}^{m}\|\nabla f_k(w_k^t; \widetilde{w}^t)\|^2 \le \mathcal{O}\left(\frac{d\gamma^2}{m}\right) + \mathcal{O}\left(\frac{\lambda B}{E}\right) + \mathcal{O}\left(\frac{1}{mE}\right)\frac{\log(1/\delta)}{\epsilon^2}. \tag{10}$$

We provide formal statement and proof of Theorem 3 in Appendix A.2. The upper bound in Equation 9 consists of two parts: error induced by the gradient descent algorithm and error induced by the Gaussian Mechanism. When $\sigma = 0$, Algorithm 1 recovers a non-private MR-MTL solver.

**Corollary 4.** *When* $\sigma = 0$, *Algorithm 1 with* $(L + \lambda)$-*smooth and nonconvex* $f_k$ *satisfies*

$$\frac{1}{mT}\sum_{t=0}^{T-1}\sum_{k=1}^{m}\|\nabla f_k(w_k^t; \widetilde{w}^t)\|^2 \le \mathcal{O}\left(\frac{\lambda}{T}\right) + \mathcal{O}\left(\frac{\lambda B}{E}\right). \tag{11}$$

By Theorem 2, given fixed $\epsilon$, $\sigma^2$ grows linearly with respect to $T$. Hence, given the same privacy guarantee, larger noise is required if the algorithm is run for more communication rounds. Note that in Theorem 3, the upper bound consists of $\mathcal{O}(\frac{1}{m\epsilon^2})$, which means when there are more tasks, the upper bound becomes smaller while the privacy parameter remains the same. On the other hand, Theorem 3 also shows a privacy-utility tradeoff using our Algorithm 1: the upper bound grows inversely proportional to the privacy parameter $\epsilon$. We also provide a convergence analysis of Algorithm 1 with strongly-convex losses in Theorem 5 below (formal statement and proof in Appendix A.3).

**Theorem 5** (Convergence under strongly-convex loss (Informal)). *Let* $f_k$ *be* $(L + \lambda)$-*smooth and* $(\mu + \lambda)$-*strongly convex. Assume* $\gamma \ge \max_{k,t}\|\nabla_{w_k^t} f_k(w_k^t; \widetilde{w}^t)\|_2$. *Let* $w_k^* = \arg\min_w f_k(w; \bar{w}^*)$ *and* $p = \frac{q}{m}$. *If we set* $\eta_t = \frac{cp}{Lp^2 + \lambda p^2 - 2\lambda}$ *for some constant* $c$ *such that* $\frac{1}{2} \le \eta p(c - 2)(\mu + \lambda) \le 1$, *we have:*

$$\frac{1}{m}\sum_{k=1}^{m} f_k(w_k^t; \widetilde{w}^t) - f_k(w_k^*; \widetilde{w}^*) \le \mathcal{O}\left(\frac{1}{2^T m}\right) + \mathcal{O}\left(\frac{2^E B}{2^E - 1}\right) + \mathcal{O}\left(\frac{2^E d\lambda\sigma^2}{2^E - 1}\right). \tag{12}$$

*Let* $\sigma$ *be chosen as in Theorem 2, then there exists* $T = \mathcal{O}\left(\frac{m}{\lambda d\gamma^2}\right)$ *such that*

$$\frac{1}{m}\sum_{k=1}^{m} f_k(w_k^t; \widetilde{w}^t) - f_k(w_k^*; \widetilde{w}^*) \le \mathcal{O}\left(\frac{1}{2^T m}\right) + \mathcal{O}\left(\frac{2^E B}{2^E - 1}\right) + \mathcal{O}\left(\frac{2^E}{m(2^E - 1)}\right)\frac{\log(1/\delta)}{\epsilon^2} \tag{13}$$

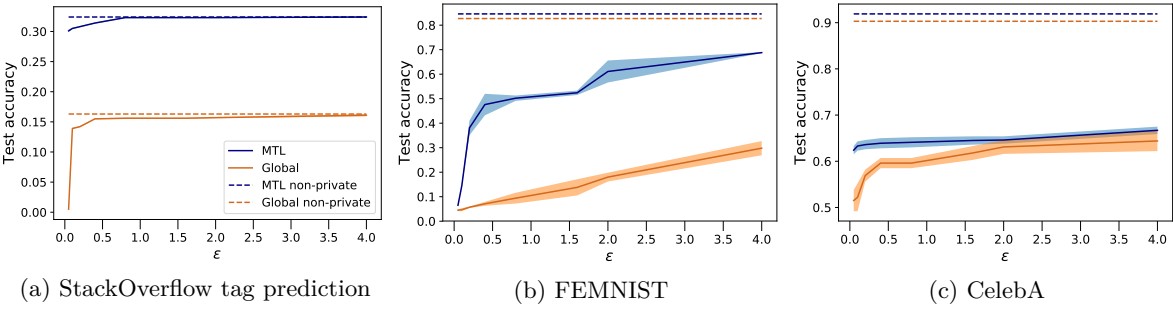

(a) StackOverflow tag prediction         (b) FEMNIST           (c) CelebA

Figure 3: Comparison of training PMTL vs. a private global model. PMTL is able to retain advantages over global approaches in private settings. In addition, even in settings where the non-private MTL and global baselines are similar (e.g., FEMNIST, CelebA), there exist utility benefits at all levels of $\epsilon$ when using PMTL.

As with Corollary 4, we recover the bound of the non-private mean-regularized MTL solver for $\sigma=0$. We provide convergence under convex loss in non-private scenario in Appendix A.3.

**Convergence to a neighborhood of the optimal.** It is worth noting that for both convex and non-convex case, given the nature of the mean-regularized MTL objective, the local objective will only converge to within a neighborhood of the optimal. Consider the following simple mean-estimation problem as an example. Assume that we have $m$ different clients/tasks, each with local data $x_i$ and local model $w_i$. The mean-regularized MTL objective for this problem would be $\frac{1}{m}\sum_i(x_i - w_i)^2 + \frac{1}{m}\sum_i(w_i - \bar{w})^2$, which is greater than $\frac{1}{2m}\sum_i(x_i - w_i + w_i - \bar{w})^2 = \frac{1}{2m}\sum_i(x_i - \bar{w})^2$. Note that this lower bound neither converges to 0 as $\bar{w}$ changes over time nor diminishes with increasing $m$. This result is therefore expected/standard, and is in line with other previous works that study the same objective but with different solvers, where similar dependencies on $\eta$ and $B$ can be seen (Hanzely & Richtárik, 2020).

## 5 Experiments

We empirically evaluate our private MTL solver on common federated learning benchmarks Caldas et al. (2018). We first demonstrate the superior privacy-utility trade-off that exists when training our private MTL method compared with training a single global model (Section 5.2). We also compare our method with simple finetuning—exploring the results of performing local finetuning after learning an MTL objective vs. a global objective (Section 5.3). Our code is publicly available at: `https://github.com/s-huu/PMTL`

### 5.1 Setup

For all experiments, we evaluate the test accuracy and privacy parameter of our private MTL solver given a fixed clipping bound $\gamma$, variance of Gaussian noise $\sigma^2$, and communication rounds $T$. All experiments are performed on common federated learning benchmarks as a natural application of multi-task learning. We provide a detailed description of datasets and models in Appendix A.4. Each dataset is naturally partitioned among $m$ different clients. Under such a scenario, each client can be viewed as a task and the data that a client generates is only visible locally.

| FEMNIST | $\epsilon = 0.1$ | | $\epsilon = 0.8$ | | $\epsilon = 2.0$ | | $\epsilon = \infty$ | |
|---|---|---|---|---|---|---|---|---|
| | **MTL** | **Global** | **MTL** | **Global** | **MTL** | **Global** | **MTL** | **Global** |
| Vanilla Finetuning | $\mathbf{0.645 \pm 0.013}$ | $0.606 \pm 0.017$ | $0.640 \pm 0.016$ | $\mathbf{0.648 \pm 0.017}$ | $\mathbf{0.677 \pm 0.008}$ | $0.653 \pm 0.010$ | $\mathbf{0.832 \pm 0.005}$ | $0.812 \pm 0.009$ |
| Mean-regularization | $\mathbf{0.608 \pm 0.011}$ | $0.581 \pm 0.011$ | $\mathbf{0.605 \pm 0.008}$ | $0.574 \pm 0.006$ | $\mathbf{0.656 \pm 0.009}$ | $0.633 \pm 0.003$ | $0.826 \pm 0.011$ | $\mathbf{0.839 \pm 0.006}$ |
| Symmetrized KL | $\mathbf{0.486 \pm 0.012}$ | $0.348 \pm 0.005$ | $\mathbf{0.584 \pm 0.012}$ | $0.481 \pm 0.016$ | $\mathbf{0.662 \pm 0.016}$ | $0.565 \pm 0.019$ | $\mathbf{0.839 \pm 0.006}$ | $0.829 \pm 0.015$ |
| EWC | $\mathbf{0.663 \pm 0.002}$ | $0.556 \pm 0.001$ | $0.595 \pm 0.004$ | $\mathbf{0.607 \pm 0.007}$ | $\mathbf{0.681 \pm 0.002}$ | $0.666 \pm 0.001$ | $\mathbf{0.837 \pm 0.001}$ | $0.823 \pm 0.005$ |

Table 1: Comparison of PMTL vs. a private global model with different local finetuning methods. $\epsilon = \infty$ corresponds to no noise and clipping, i.e., training non-privately. The higher accuracy between MTL and Global given the same $\epsilon$ and finetuning method is **bolded**.

### 5.2 Privacy-Utility Trade-off of PMTL

We first explore the training loss (**orange**) and privacy parameter $\epsilon$ (**blue**) as a function of communication rounds across three datasets (Figure 2). Specifically, we evaluate the average loss for all the tasks and $\epsilon$ given

a fixed $\delta$ after each round, where $\delta$ is set to be $\frac{1}{m}$ for all experiments. In general, for a fixed clipping bound $\gamma$ and $\sigma$, we see that the method converges fairly quickly with respect to the resulting privacy, but that privacy guarantees may be sacrificed in order to achieve very small losses.

To put these results in context, we also compare the test performance of our PMTL solver with that of training a global model. In particular, we use FedAvg (McMahan et al., 2017) to train a global model. At each communication round, clients solve their local objective individually. While aggregating the model updates, the global learner applies Gaussian Mechanism and sends the noisy aggregation back to the clients. As a result, private FedAvg differs from our PMTL solver in the following two places: (i) the MTL objective solved locally by each task learner has a mean-regularized term; (ii) the MTL method evaluates on one task-specific model for every task while the global method evaluates all tasks on one global model. For each dataset, we select privacy parameter $\epsilon \in [0.05, 0.1, 0.2, 0.4, 0.8, 1.6, 2.0, 4.0]$. For each $\epsilon$, we select the $\gamma$, $\sigma$, and $T$ that result in the best validation accuracy for a given $\epsilon$ and record the test accuracy. A detailed description of hyperparameters is listed in Appendix A.5. We plot the test accuracy with respect to the highest validation accuracy given one $\epsilon$ for both private MTL model and private global model. Results are shown in Figure 3.

In all three datasets, our private MTL solver achieves higher test accuracy compared with training a private global model with FedAvg given the same $\epsilon$. Moreover, the proposed mean regularized MTL solver is able to retain an advantage over global model even with noisy aggregation. In particular, for small $\epsilon < 1$, adding random Gaussian noise during global aggregation amplifies the test accuracy difference between our MTL solver and FedAvg. Under the StackOverflow task, both methods obtain test accuracy close to the non private baseline for large $\epsilon$. To demonstrate that applying private MTL has an advantage over private global training more generally, we also compared our PMTL method with private FedProx (Li et al., 2020b). The results (which mirror Figure 3) are in Appendix A.6.

### 5.3 PMTL with local finetuning

Finally, in federated learning, previous works have shown local finetuning with different objectives is helpful for improving utility while training a differentially private global model (Yu et al., 2020). In this section, after obtaining a private global model, we explore locally finetuning the task specific models by optimizing different local objective functions. In particular, we use common objectives which (i) naively optimize the local empirical risk (Vanilla Finetuning), or (ii) encourage minimizing the distance between local and global model under different distance metrics (Mean-regularization, Symmetrized KL, EWC (Kirkpatrick et al., 2017; Yu et al., 2020)). The results are listed in Table 1. When $\epsilon = \infty$ (the non-private setting), global with mean-regularization finetuning outperforms all MTL+finetuning methods. However, when we add privacy to both methods, private MTL+finetuning has an advantage over global with finetuning on different finetuning objectives. In some cases, e.g. using Symmetrized KL, the test accuracy gap between private MTL+fintuning and private global+finetuning is amplified when $\epsilon$ is small compared to the case where no privacy is added during training. We also compare our PMTL+finetuning with training pure local model in Appendix A.7.

## 6 Conclusion and Future work

In this work, we defined notions of client-level differential privacy for multi-task learning and proposed a simple method for private mean-regularized MTL. Theoretically, we provided both privacy and utility guarantees for our approach. Empirically, we showed that private MTL retains advantages over training a private global model on common federated learning benchmarks. In future work, we are interested in building on our results to explore privacy for more general forms of MTL, e.g., the family of objectives in (1) with arbitrary $\Omega$, as well as studying how client-level privacy relates to issues of fairness in multi-task settings.

## 7 Acknowledgements

ZSW was supported in part by the NSF FAI Award #1939606, NSF Award #2120667, a Google Faculty Research Award, a J.P. Morgan Faculty Award, a Meta Research Award, and a Cisco research grant.

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

# A  Appendix

## A.1  Privacy Analysis: Proof for Theorem 1 and 2

In the proofs of Theorem 1 and 2 we follow the line of reasoning in Abadi et al. (2016), which analyzes the privacy of DPSGD. We first state the following lemma from Abadi et al. (2016).

**Lemma 1.** *(Abadi et al., 2016, Theorem 1)  There exists constants $c_1$ and $c_2$ such that given the sampling probability $p = \frac{q}{m}$ and the number of steps $T$, for any $\epsilon < c_1 p^2 T$, DPSGD is $(\epsilon, \delta)$-differentially private for any $\delta > 0$ if we choose $\sigma \geq c_2 \frac{p\sqrt{T \log(1/\delta)}}{\epsilon}$.*

To prove Theorem 1, we also need the following definitions and lemmas.

**Definition 4** ($\ell_2$-sensitivity)**.** *Let $f : \mathcal{U} \to \mathbb{R}^d$ be some arbitrary function, the $\ell_2$-sensitivity of $f$ is defined as*

$$\Delta_2 f = \max_{adjacent\ D, D' \in \mathcal{U}} \|f(D) - f(D')\|_2 \tag{14}$$

**Definition 5** (Rényi Divergence)**.** *(Mironov, 2017, Definition 3) Let $P, Q$ be two probability distribution over the same probability space, and let $p, q$ be the respective probability density function. The Rényi Divergence with finite order $\alpha \neq 1$ is:*

$$D_\alpha(P\|Q) = \frac{1}{\alpha - 1} \ln \int_{\mathcal{X}} q(x) \left( \frac{p(x)}{q(x)} \right)^\alpha dx \tag{15}$$

**Definition 6** ($(\alpha, \epsilon)$-Rényi Differential Privacy)**.** *(Mironov, 2017, Definition 4) A randomized mechanism $f : \mathcal{D} \to \mathcal{R}$ is said to have $(\alpha, \epsilon)$-Rényi Differential Privacy if for all adjacent $D, D' \in \mathcal{D}$ it holds that:*

$$D_\alpha(f(D)\|f(D')) \leq \epsilon. \tag{16}$$

**Lemma 2.** *(Mironov, 2017, Corollary 3)  The Gaussian mechanism is $(\alpha, \alpha(2(\Delta_2 f)^2/\sigma^2))$-Renyi Differentially Private.*

**Lemma 3.** *(Mironov, 2017, Proposition 3)  If $f$ is $(\alpha, \epsilon)$-RDP, then it is $(\epsilon + \frac{\log(1/\delta)}{\alpha - 1}, \delta)$-DP for all $\delta > 0$.*

We begin by proving the first part of Theorem 1, where $q \neq m$.

*Proof for Theorem 1: $q \neq m$.* Note that aggregation step in line 8 of Algorithm 1 can be rewritten as

$$\widetilde{w}^{t+1} = \widetilde{w}^t + \frac{1}{|S_t|} \sum_{k \in S_t} g_k^{t+1} \min \left( 1, \frac{\gamma}{\|g_k^{t+1}\|_2} \right) + \mathcal{N}(0, \sigma^2 \mathbf{I}_{\mathbf{d} \times \mathbf{d}}) \tag{17}$$

$$= \widetilde{w}^t + \frac{1}{q} \sum_{k \in S_t} g_k^{t+1} \min \left( 1, \frac{\gamma}{\|g_k^{t+1}\|_2} \right) + \mathcal{N}\left( 0, \left( \frac{\sigma}{\gamma} \right)^2 \gamma^2 \mathbf{I}_{\mathbf{d} \times \mathbf{d}} \right) \tag{18}$$

$$= \widetilde{w}^t + \frac{1}{q} \left( \sum_{k \in S_t} g_k^{t+1} \min \left( 1, \frac{\gamma}{\|g_k^{t+1}\|_2} \right) + \mathcal{N}\left( 0, \left( \frac{q\sigma}{\gamma} \right)^2 \gamma^2 \mathbf{I}_{\mathbf{d} \times \mathbf{d}} \right) \right). \tag{19}$$

From here, we can directly apply Lemma 1 with $\sigma$ set to be $\frac{q\sigma}{\gamma}$. Hence, we conclude that when $q \neq m$, there exists constants $c_1$ and $c_2$ such that given the number of steps $T$, for any $\epsilon < c_1 \frac{q^2}{m^2} T$, $\mathcal{M}^{1:T}$ is $(\epsilon, \delta)$-differentially private for any $\delta > 0$ if we choose $\sigma \geq c_2 \frac{\gamma\sqrt{T \log(1/\delta)}}{m\epsilon}$. $\qquad\square$

This proof can extend to the case where $q = m$. In the remainder of this section, we provide a proof that gives a more specific bound on the variance $\sigma^2$ in the case where $q = m$.

*Proof for Theorem 1: $q = m$.* Define $H^t : \prod_{i=1}^m \mathcal{D}_i \times \mathcal{W} \to \mathcal{W}$ as

$$H^t(\{D_i\}, \{h_i(\cdot)\}, \widetilde{w}^t) = \widetilde{w}^t + \frac{1}{m} \sum_{i=1}^m h_i^t(D_i, \widetilde{w}^t). \tag{20}$$

As a result, we have $\mathcal{M}^t(\{D_i\}, \{h_i(\cdot)\}, \widetilde{w}^t, \sigma) = H^t(\{D_i\}, \{h_i(\cdot)\}, \widetilde{w}^t) + \beta^t$.

By Lemma 2, $\mathcal{M}^t$ is $(\alpha, 2\alpha(\Delta_2 H^t)^2/d\sigma^2)$-Renyi Differentially Private. Note that

$$(\Delta_2 H^t)^2 = \max_j \max_{\text{adjacent } D_j, D'_j \in \mathcal{D}_j} \left\| H^t(\{D_1, \cdots, D_j, \cdots, D_m\}) - H^t(\{D_1, \cdots, D'_j, \cdots, D_m\}) \right\|^2 \quad (21)$$

$$= \max_j \max_{\text{adjacent } D_j, D'_j \in \mathcal{D}_j} \left\| \frac{1}{m} h_j^t(D_j, \widetilde{w}^t) - \frac{1}{m} h_j^t(D'_j, \widetilde{w}^t) \right\|^2 \quad (22)$$

$$= \frac{1}{m^2} \max_j \max_{\text{adjacent } D_j, D'_j \in \mathcal{D}_j} \left\| h_j^t(D_j, \widetilde{w}^t) - h_j^t(D'_j, \widetilde{w}^t) \right\|^2 \quad (23)$$

$$= \frac{1}{m^2} \max_j (\Delta_2 h_j^t)^2. \quad (24)$$

Hence, by sequential composition of Rényi Differential Privacy (Mironov, 2017, Proposition 1), $\mathcal{M}^{1:T}$ is $(\alpha, \sum_{i=1}^{T} 2\alpha \max_j (\Delta_2 h_j^t)^2/m^2\sigma^2)$-RDP.

By Lemma 3, we know that $\mathcal{M}^{1:T}$ is $(\sum_{i=1}^{T} 2\alpha \max_j (\Delta_2 h_j^t)^2/m^2\sigma^2 + \frac{\log(1/\delta)}{\alpha-1}, \delta)$-DP.

Plugging in $\alpha = \frac{4\log(1/\delta)}{\epsilon}$, $\sigma = \frac{4\gamma\sqrt{T\log(1/\delta)}}{\epsilon m}$, we have

$$\sum_{i=1}^{T} 2\alpha \max_j (\Delta_2 h_j^t)^2/m^2\sigma^2 + \frac{\log(1/\delta)}{\alpha-1} \le \sum_{i=1}^{T} 2\alpha\gamma^2/m^2\sigma^2 + \frac{\log(1/\delta)}{\alpha-1} \quad (25)$$

$$= \frac{2\frac{4\log(1/\delta)}{\epsilon}\gamma^2}{m^2(\frac{4\gamma\sqrt{T\log(1/\delta)}}{\epsilon m})^2} + \frac{\log(1/\delta)}{\frac{4\log(1/\delta)}{\epsilon} - 1} \quad (26)$$

$$\le \frac{\epsilon}{2} + \frac{\epsilon}{2} \quad (27)$$

$$= \epsilon. \quad (28)$$

Hence, $\mathcal{M}^{1:T}$ is $(\epsilon, \delta)$-DP if we choose $\sigma = \frac{4\gamma\sqrt{T\log(1/\delta)}}{\epsilon m}$. $\qquad \square$

By Theorem 1 and *Billboard Lemma*, it directly follows that Algorithm 1 is $(\epsilon, \delta)-$JDP.

*Proof for Theorem 2.* Theorem 1 shows that Algorithm 1 consists of a $(\epsilon, \delta)$-DP process to produce global model. After that each task learner trains local model with the DP global model and its private data. By definition of BP, it directly follows that Algorithm 1 is $(\epsilon, \delta)$-BP. $\qquad \square$

## A.2  Convergence Analysis(nonconvex):

We first present the formal statement of Theorem 3.

**Theorem 6** (Convergence under nonconvex loss). *Let $f_k$ be $(L+\lambda)$-smooth. Assume $\gamma$ is sufficiently large such that $\gamma \ge \max_{k,t} \|\nabla_{w_k^t} f_k(w_k^t; \widetilde{w}^t)\|_2$. Further let $f_k^* = \min_{w,\bar{w}} f_k(w; \bar{w})$ and $p = \frac{q}{m}$. If we use a fixed learning rate $\eta_t = \eta = \frac{1}{pL+(p+\frac{1}{p})\lambda}$, Algorithm 1 satisfies:*

$$\frac{1}{mT} \sum_{t=0}^{T-1} \sum_{k=1}^{m} \|\nabla f_k(w_k^t; \widetilde{w}^t)\|^2 \le \frac{\left(4\left(L+\lambda+\frac{1}{p^2}\lambda\right) + \frac{2\lambda}{E(Lp^2+\lambda p^2+2\lambda)}\right) \sum_{k=1}^{m} (f_k(w_k^0; \widetilde{w}^0) - f_k^*)}{mT}$$

$$+ \frac{\mathcal{O}\left(L+\lambda+\frac{\lambda}{p^2}\right) \sum_{i=1}^{T/E} B_{iE}}{T} + \mathcal{O}\left(\frac{Ld\lambda + d\lambda^2 + d\lambda^2/p^2}{mE}\right) \sigma^2. \quad (29)$$

*where*

$$B_t = \max_k f_k(w_k^t; \widetilde{w}^t). \quad (30)$$

*Let $\sigma$ chosen as we set in Theorem 2. Take $T = \mathcal{O}\left(\frac{m}{\lambda d \gamma^2}\right)$, the right hand side is bounded by*

$$\frac{1}{mT}\sum_{t=0}^{T-1}\sum_{k=1}^{m}\|\nabla f_k(w_k^t; \widetilde{w}^t)\|^2 \leq \frac{\mathcal{O}\left(4\left(L + \lambda + \frac{1}{p^2}\lambda\right) + \frac{2\lambda}{E(Lp^2 + \lambda p^2 + 2\lambda)}\right)}{m}$$

$$+ \mathcal{O}\left(\frac{L + \lambda + \frac{\lambda}{p^2}}{E}\right)B + \mathcal{O}\left(\frac{L + \lambda + \frac{\lambda}{p^2}}{mE}\right)\frac{\log(1/\delta)}{\epsilon^2}. \tag{31}$$

*Proof for Theorem 6.* Let $w_k^* = \arg\min_w f_k(w; \bar{w}^*)$. Let $I_k^t$ be the random variable indicating whether task $k$ is selected in communication round $t$. Note that the probability task learner $k$ is selected in any arbitrary communication round $p_k = \dfrac{\binom{m-1}{q-1}}{\binom{m}{q}} = \frac{q}{m}$. Thus $\mathbb{E}[I_k^t] = p_k = \frac{q}{m}$. By $L$-smoothness of $f_k$, we have

$$\mathbb{E}[f_k(w_k^{t+1}; \widetilde{w}^t) - f_k(w_k^t; \widetilde{w}^t)] \leq \mathbb{E}\left[\langle \nabla f_k(w_k^t; \widetilde{w}^t), w_k^{t+1} - w_k^t\rangle + \frac{L}{2}\|w_k^{t+1} - w_k^t\|^2\right] \tag{32}$$

$$= \mathbb{E}\left[\langle \nabla f_k(w_k^t; \widetilde{w}^t), \eta_t I_k^t \nabla f_k(w_k^t; \widetilde{w}^t)\rangle + \frac{L}{2}\|\eta_t I_k^t \nabla f_k(w_k^t; \widetilde{w}^t)\|^2\right] \tag{33}$$

$$= \left(\frac{L+\lambda}{2}\eta_t^2 p_k^2 - \eta_t p_k\right)\|\nabla f_k(w_k^t; \widetilde{w}^t)\|^2. \tag{34}$$

In the case where $t + 1 \not\equiv 0 \mod E$, i.e. $t+1$ is not a communication round, $\widetilde{w}^{t+1} = \widetilde{w}^t$. Therefore, we have

$$\mathbb{E}[f_k(w_k^{t+1}; \widetilde{w}^{t+1}) - f_k(w_k^t; \widetilde{w}^t)] \leq \left(\frac{L+\lambda}{2}\eta_t^2 p_k^2 - \eta_t p_k\right)\|\nabla f_k(w_k^t; \widetilde{w}^t)\|^2. \tag{35}$$

In the case where $t + 1 \equiv 0 \mod E$, we have

$$\mathbb{E}[f_k(w_k^{t+1}; \widetilde{w}^{t+1}) - f_k(w_k^t; \widetilde{w}^t)] \leq \underbrace{\mathbb{E}[f_k(w_k^{t+1}; \widetilde{w}^{t+1}) - f_k(w_k^{t+1}; \widetilde{w}^t)]}_{B} + \left(\frac{L+\lambda}{2}\eta_t^2 p_k^2 - \eta_t p_k\right)\|\nabla f_k(w_k^t; \widetilde{w}^t)\|^2. \tag{36}$$

It suffices to bound B:

$$B = \mathbb{E}\left[\frac{\lambda}{2}\|w_k^{t+1} - \widetilde{w}^{t+1}\|^2 - \frac{\lambda}{2}\|w_k^{t+1} - \widetilde{w}^t\|^2\right] \tag{37}$$

$$= \frac{\lambda}{2}\mathbb{E}[\|\widetilde{w}^t - \widetilde{w}^{t+1}\|\|2w_k^{t+1} - \widetilde{w}^t - \widetilde{w}^{t+1}\|] \tag{38}$$

$$\leq \frac{\lambda}{2}\sqrt{\mathbb{E}[\|\widetilde{w}^t - \widetilde{w}^{t+1}\|^2]\mathbb{E}[\|2w_k^{t+1} - \widetilde{w}^t - \widetilde{w}^{t+1}\|^2]} \tag{39}$$

$$= \frac{\lambda}{2}\sqrt{\mathbb{E}[\|\widetilde{w}^t - \widetilde{w}^{t+1}\|^2]}\sqrt{\mathbb{E}[\|(\widetilde{w}^{t+1} - \widetilde{w}^t) + 2(w_k^{t+1} - \widetilde{w}^{t+1})\|^2]} \tag{40}$$

$$\leq \frac{\lambda}{2}\sqrt{\mathbb{E}[\|\widetilde{w}^t - \widetilde{w}^{t+1}\|^2]}\sqrt{\mathbb{E}[\|\widetilde{w}^{t+1} - \widetilde{w}^t\|^2] + 4\|w_k^{t+1} - \widetilde{w}^{t+1}\|^2 + 4\mathbb{E}[\|\widetilde{w}^{t+1} - \widetilde{w}^t\|\|w_k^{t+1} - \widetilde{w}^{t+1}\|]} \tag{41}$$

$$\leq \frac{\lambda}{2}\sqrt{\underbrace{\mathbb{E}[\|\widetilde{w}^t - \widetilde{w}^{t+1}\|^2]}_{C_1}}\sqrt{\mathbb{E}[\|\widetilde{w}^{t+1} - \widetilde{w}^t\|^2] + 4\underbrace{\|w_k^{t+1} - \widetilde{w}^{t+1}\|^2}_{C_2} + 4\sqrt{\mathbb{E}[\|\widetilde{w}^{t+1} - \widetilde{w}^t\|^2]\|w_k^{t+1} - \widetilde{w}^{t+1}\|^2}} \tag{42}$$

where the first and third inequality follows from Cauchy-Schwartz Inequality: $\mathbb{E}[XY] \le \sqrt{\mathbb{E}[X^2]\mathbb{E}[Y^2]}$. We can then upper bound $C_1$ and $C_2$.

$$C_1 = \mathbb{E}\left[\left\|\frac{1}{q}\sum_{k\in S_t} g_k^{t+1}\min\left(1, \frac{\gamma}{\|g_k^{t+1}\|_2}\right) + \beta^t\right\|^2\right] \tag{43}$$

$$\le \left(\sqrt{\mathbb{E}\left[\left\|\frac{1}{q}\sum_{k=1}^m I_k^{t+1}g_k^{t+1}\min\left(1, \frac{\gamma}{\|g_k^{t+1}\|_2}\right)\right\|^2\right]} + \sqrt{\mathbb{E}[\|\beta^t\|^2]}\right)^2 \tag{44}$$

$$= \left(\sqrt{\mathbb{E}\left[\left\|\frac{1}{q}\sum_{k=1}^m I_k^{t+1}g_k^{t+1}\min\left(1, \frac{\gamma}{\|g_k^{t+1}\|_2}\right)\right\|^2\right]} + \sqrt{d}\sigma\right)^2 \tag{45}$$

$$\le \left(\sqrt{\frac{m}{q^2}\sum_{k=1}^m \mathbb{E}\left[\left\|I_k^{t+1}g_k^{t+1}\min\left(1, \frac{\gamma}{\|g_k^{t+1}\|_2}\right)\right\|^2\right]} + \sqrt{d}\sigma\right)^2 \tag{46}$$

$$\le \left(\sqrt{\frac{m}{q^2}\sum_{k=1}^m \mathbb{E}\left[\left\|I_k^{t+1}\eta_t\nabla f_k(w_k^t)\min\left(1, \frac{\gamma}{\eta_t\|\nabla f_k(w_k^t)\|_2}\right)\right\|^2\right]} + \sqrt{d}\sigma\right)^2 \tag{47}$$

$$\le \left(\sqrt{\frac{1}{m}\sum_{k=1}^m \left\|\eta_t\nabla f_k(w_k^t)\min\left(1, \frac{\gamma}{\eta_t\|\nabla f_k(w_k^t)\|_2}\right)\right\|^2} + \sqrt{d}\sigma\right)^2. \tag{48}$$

Denote $h(t) = \sqrt{\frac{1}{m}\sum_{k=1}^m \left\|\eta_t\nabla f_k(w_k^t)\min\left(1, \frac{\gamma}{\eta_t\|\nabla f_k(w_k^t)\|_2}\right)\right\|^2}$. We have:

$$C_2 \le \frac{2}{\lambda}\frac{\lambda}{2}\|w_k^{t+1} - \widetilde{w}^{t+1}\|^2 \tag{49}$$

$$\le \frac{2}{\lambda}f_k(w_k^{t+1}; \widetilde{w}^{t+1}) \tag{50}$$

$$= \frac{2}{\lambda}B_{t+1} \tag{51}$$

Plugging the bounds for $C_1$ and $C_2$ into B yields:

$$B \le \frac{\lambda}{2}(h(t) + \sqrt{d}\sigma)\left(h(t) + \sqrt{d}\sigma + 2\sqrt{\frac{2}{\lambda}B_{t+1}}\right). \tag{52}$$

Denote the right hand side as $\beta(t)$, we have

$$\mathbb{E}[f_k(w_k^{t+1}; \widetilde{w}^{t+1}) - f_k(w_k^t; \widetilde{w}^t)] \le \beta(t) + \left(\frac{L+\lambda}{2}\eta_t^2 p_k^2 - \eta_t p_k\right)\|\nabla f_k(w_k^t; \widetilde{w}^t)\|^2. \tag{53}$$

Let $\delta_t = \mathbb{E}[f_k(w_k^t; \widetilde{w}^t) - f_k(w_k^*; \overline{w}^*)]$, we have

$$\delta_{t+1} \le \delta_t + \beta(t) + \left(\frac{L+\lambda}{2}\eta_t^2 p_k^2 - \eta_t p_k\right)\|\nabla f_k(w_k^t; \widetilde{w}^t)\|^2. \tag{54}$$

In the nonconvex case, we have

$$\sum_{t=0}^{T-1}\left(\eta_t p_k - \frac{L+\lambda}{2}\eta_t^2 p_k^2\right)\|\nabla f_k(w_k^t; \widetilde{w}^t)\|^2 - \beta(t) \le f_k(w_k^0; \widetilde{w}^0) - f_k^* \tag{55}$$

Summing over $k$ on the left handed side, when $\gamma$ is large enough so that no clipping happens we have

$$\sum_{k=1}^{m}\sum_{t+1\equiv 0\mod E}\left(\left(\eta_t p_k - \frac{L+\lambda}{2}\eta_t^2 p_k^2\right)\|\nabla f_k(w_k^t;\widetilde{w}^t)\|^2 - \beta(t)\right)$$
$$+\sum_{t+1\not\equiv 0\mod E}\left(\eta_t p_k - \frac{L+\lambda}{2}\eta_t^2 p_k^2\right)\|\nabla f_k(w_k^t;\widetilde{w}^t)\|^2 \tag{56}$$

$$=\sum_{t+1\equiv 0\mod E}\left(\eta_t p - \frac{L+\lambda}{2}\eta_t^2 p^2\right)\sum_{k=1}^{m}\|\nabla f_k(w_k^t;\widetilde{w}^t)\|^2 - m\beta(t)$$
$$+\sum_{t+1\not\equiv 0\mod E}\left(\eta_t p - \frac{L+\lambda}{2}\eta_t^2 p^2\right)\sum_{k=1}^{m}\|\nabla f_k(w_k^t;\widetilde{w}^t)\|^2 \tag{57}$$

$$\sum_{t+1\equiv 0\mod E}\left(\eta_t p - \frac{L+\lambda}{2}\eta_t^2 p^2\right)\sum_{k=1}^{m}\|\nabla f_k(w_k^t;\widetilde{w}^t)\|^2$$
$$=-\frac{\lambda}{2}\left(mh^2(t) + \left(2\sqrt{d}\sigma + 2\sqrt{\frac{2}{\lambda}B_{t+1}}\right)mh(t) + m\left(d\sigma^2 + 2\sigma\sqrt{\frac{2d}{\lambda}B_{t+1}}\right)\right) \tag{58}$$
$$+\sum_{t+1\not\equiv 0\mod E}\left(\eta_t p - \frac{L+\lambda}{2}\eta_t^2 p^2\right)\sum_{k=1}^{m}\|\nabla f_k(w_k^t;\widetilde{w}^t)\|^2$$

$$\sum_{t=0}^{T-1}\left(\eta_t p - \frac{L+\lambda}{2}\eta_t^2 p^2\right)G_t^2$$
$$=+\sum_{t+1\not\equiv 0\mod E}-\frac{\lambda}{2}\eta_t^2 G_t^2 - \lambda\sqrt{m}\left(\sqrt{d}\sigma + \sqrt{\frac{2}{\lambda}B_{t+1}}\right)\eta_t G_t - \frac{\lambda m}{2}\left(d\sigma^2 + 2\sigma\sqrt{\frac{2d}{\lambda}B_{t+1}}\right) \tag{59}$$

$$\leq \sum_{k=1}^{m} f_k(w_k^0;\widetilde{w}^0) - f_k^*, \tag{60}$$

where $G_t = \sqrt{\sum_{k=1}^{m}\|\nabla f_k(w_k^t;\widetilde{w}^t)\|^2}$. Picking $\eta_t = \frac{p}{p^2 L + (p^2+1)\lambda}$ yields

$$\sum_{t+1\equiv 0\mod E}\frac{p^2}{2(p^2 L + (p^2+1)\lambda)}G_t^2 - \frac{\lambda\sqrt{m}\left(\sqrt{d}\sigma + \sqrt{\frac{2}{\lambda}B_{t+1}}\right)p}{p^2 L + (p^2+1)\lambda}G_t - \frac{\lambda m}{2}\left(d\sigma^2 + 2\sigma\sqrt{\frac{2d}{\lambda}B_{t+1}}\right) \tag{61}$$

$$+\sum_{t+1\not\equiv 0\mod E}\frac{p^2(p^2 L + (p^2+2)\lambda)}{2(p^2 L + (p^2+1)\lambda)^2}G_t^2 \tag{62}$$

$$\leq \sum_{k=1}^{m} f_k(w_k^0;\widetilde{w}^0) - f_k^*. \tag{63}$$

This is equivalent to

$$\sum_{t=0}^{T-1}G_t^2 + \sum_{t+1\equiv 0\mod E}-2\lambda\sqrt{m}\left(\sqrt{d}\sigma + \sqrt{\frac{2}{\lambda}B_{t+1}}\right)\frac{m}{q}G_t - \left(L+\lambda+\frac{\lambda}{p^2}\right)\lambda m\left(d\sigma^2 + 2\sigma\sqrt{\frac{2d}{\lambda}B_{t+1}}\right) \tag{64}$$

$$\leq \left(2\frac{1}{E}\left(L+\lambda+\frac{1}{p^2}\lambda\right) + 2\frac{E-1}{E}\frac{\left(L+\lambda+\frac{1}{p^2}\lambda\right)^2}{L+\lambda+\frac{2}{p^2}\lambda}\right)\sum_{k=1}^{m} f_k(w_k^0;\widetilde{w}^0) - f_k^* \tag{65}$$

$$=\left(2\left(L+\lambda+\frac{1}{p^2}\lambda\right) + \frac{\frac{1}{p^2}\lambda}{E\left(L+\lambda+\frac{2}{p^2}\lambda\right)}\right)\sum_{k=1}^{m} f_k(w_k^0;\widetilde{w}^0) - f_k^*. \tag{66}$$

Hence, we have

$$\sum_{t+1\equiv 0 \mod E} \left(G_t - \lambda m \left(\sqrt{d}\sigma + \sqrt{\frac{2}{\lambda}B_{t+1}}\right)\frac{1}{p}\right)^2 + \sum_{t+1\not\equiv 0 \mod E} G_t^2 \tag{67}$$

$$\leq \left(2\left(L + \lambda + \frac{1}{p^2}\lambda\right) + \frac{\frac{1}{p^2}\lambda}{E\left(L + \lambda + \frac{2}{p^2}\lambda\right)}\right)\sum_{k=1}^{m} f_k(w_k^0;\widetilde{w}^0) - f_k^* \tag{68}$$

$$+ \sum_{t+1\equiv 0 \mod E} (L\lambda m + m\lambda^2)\left(d\sigma^2 + 2\sigma\sqrt{\frac{2d}{\lambda}B_{t+1}}\right) + 2\frac{m\lambda B_{t+1}}{p^2}. \tag{69}$$

This implies

$$\sum_{t=0}^{T-1} G_t^2 \leq 2\left(2\left(L + \lambda + \frac{1}{p^2}\lambda\right) + \frac{\frac{1}{p^2}\lambda}{E\left(L + \lambda + \frac{2}{p^2}\lambda\right)}\right)\sum_{k=1}^{m} f_k(w_k^0;\widetilde{w}^0) - f_k^* \tag{70}$$

$$+ 2\left(\sum_{t+1\equiv 0 \mod E} (L\lambda m + m\lambda^2 + \frac{2\lambda^2 m}{p^2})\left(d\sigma^2 + 2\sigma\sqrt{\frac{2d}{\lambda}B_{t+1}}\right) + 4\frac{m\lambda B_{t+1}}{p^2}\right). \tag{71}$$

Hence, we conclude that

$$\frac{1}{mT}\sum_{t=0}^{T-1}\sum_{k=1}^{m}\|\nabla f_k(w_k^t;\widetilde{w}^t)\|^2 \tag{72}$$

$$\leq \frac{\left(4\left(L + \lambda + \frac{1}{p^2}\lambda\right) + \frac{2\lambda}{E(Lp^2 + \lambda p^2 + 2\lambda)}\right)\sum_{k=1}^{m}(f_k(w_k^0;\widetilde{w}^0) - f_k^*)}{mT}$$
$$+ \frac{\mathcal{O}\left(L\lambda + \lambda^2 + \frac{\lambda^2}{p^2}\right)\sum_{i=1}^{T/E}\left(d\sigma^2 + 2\sigma\sqrt{\frac{2d}{\lambda}B_{iE}} + \frac{2B_{iE}}{\lambda}\right)}{T} \tag{73}$$

$$\leq \frac{\left(4\left(L + \lambda + \frac{1}{p^2}\lambda\right) + \frac{2\lambda}{E(Lp^2 + \lambda p^2 + 2\lambda)}\right)\sum_{k=1}^{m}(f_k(w_k^0;\widetilde{w}^0) - f_k^*)}{mT}$$
$$+ \frac{\mathcal{O}\left(L + \lambda + \frac{\lambda}{p^2}\right)\sum_{i=1}^{T/E}\left(\sqrt{d\lambda}\sigma + \sqrt{2B_{iE}}\right)^2}{T} \tag{74}$$

$$\leq \frac{\left(4\left(L + \lambda + \frac{1}{p^2}\lambda\right) + \frac{2\lambda}{E(Lp^2 + \lambda p^2 + 2\lambda)}\right)\sum_{k=1}^{m}(f_k(w_k^0;\widetilde{w}^0) - f_k^*)}{mT} + \frac{\mathcal{O}\left(L + \lambda + \frac{\lambda m^2}{q^2}\right)\sum_{i=1}^{T/E}B_{iE}}{T}$$
$$+ \mathcal{O}\left(\frac{Ld\lambda + d\lambda^2 + d\lambda^2/p^2}{E}\right)\sigma^2. \tag{75}$$

Taking $\sigma = \frac{c_2\gamma\sqrt{T\log(1/\delta)}}{m\epsilon}$ and $T = \mathcal{O}\left(\frac{m}{\lambda d\gamma^2}\right)$, we have

$$\frac{1}{mT}\sum_{t=0}^{T-1}\sum_{k=1}^{m}\|\nabla f_k(w_k^t;\widetilde{w}^t)\|^2 \leq \frac{\left(4\left(L + \lambda + \frac{1}{p^2}\lambda\right) + \frac{2\lambda}{E(Lp^2 + \lambda p^2 + 2\lambda)}\right)\sum_{k=1}^{m}(f_k(w_k^0;\widetilde{w}^0) - f_k^*)}{mT} \tag{76}$$

$$+ \frac{\mathcal{O}\left(L + \lambda + \frac{\lambda}{p^2}\right)\sum_{t=0}^{T-1}B_{t+1}}{T} + \frac{1}{E}\mathcal{O}\left(L + \lambda + \frac{\lambda}{p^2}\right)\frac{\log(1/\delta)}{\epsilon^2}. \tag{77}$$

$\square$

### A.3 Convergence Analysis (Convex):

We first present the formal statement of Theorem 5.

**Theorem 7.** *Let $f_k$ be $(L + \lambda)$-smooth and $(\mu + \lambda)$-strongly convex. Assume $\gamma$ is sufficiently large such that $\gamma \geq \max_{k,t} \|\nabla_{w_k^t} f_k(w_k^t; \widetilde{w}^t)\|_2$. Further let $w_k^* = \arg\min_w f_k(w; \bar{w}^*)$, where $\bar{w}^* = \frac{1}{m} \sum_{k=1}^m w_k^*$ and $p = \frac{q}{m}$. If we use a fixed learning rate $\eta_t = \eta = \frac{cp}{Lp^2 + \lambda p^2 - 2\lambda}$ for some constant $c$ such that $0 \leq 1 - \eta p(c - 2)(\mu + \lambda) \leq \frac{1}{2}$, Algorithm 1 satisfies:*

$$\Delta_T \leq \frac{1}{2^T}\left(\Delta_0 - m\lambda\left(d\sigma^2 + 2\sqrt{d}\sigma\sqrt{\frac{2}{\lambda}B} + \frac{1}{\lambda}B\right)\right) + \frac{m\lambda\left(d\sigma^2 + 2\sqrt{d}\sigma\sqrt{\frac{2}{\lambda}B} + \frac{1}{\lambda}B\right)}{1 - \frac{1}{2^E}}, \tag{78}$$

*where $\Delta_t = \sum_{k=1}^m f_k(w_k^t; \widetilde{w}^t) - f_k(w_k^*; \widetilde{w}^*)$ and $B = \max_t \max_k f_k(w_k^t; \widetilde{w}^t)$.*

*Let $\sigma$ be chosen as in Theorem 2, then there exists $T = \mathcal{O}\left(\frac{m}{\lambda d\gamma^2}\right)$ such that*

$$\frac{1}{m}\Delta_T \leq \frac{1}{2^T}\left(\frac{1}{m}\Delta_0 - \frac{\log(1/\delta)}{m\epsilon^2} - \mathcal{O}(B)\right) + \frac{\frac{\log(1/\delta)}{m\epsilon^2} + \mathcal{O}(B)}{1 - \frac{1}{2^E}}. \tag{79}$$

*Proof for Theorem 7.* Let $w_k^* = \arg\min_w f_k(w; \bar{w}^*)$. Let $I_k^t$ be the random variable indicating whether task $k$ is selected in communication round $t$. Thus $\mathbb{E}[I_k^t] = p_k$. By $L + \lambda$-smoothness and $\mu + \lambda$-strong convexity of $f_k$, we have

$$\mathbb{E}[f_k(w_k^{t+1}; \widetilde{w}^t) - f_k(w_k^t; \widetilde{w}^t)] \leq \mathbb{E}\left[\langle\nabla f_k(w_k^t; \widetilde{w}^t), w_k^{t+1} - w_k^t\rangle + \frac{L}{2}\|w_k^{t+1} - w_k^t\|^2\right] \tag{80}$$

$$= \mathbb{E}\left[\langle\nabla f_k(w_k^t; \widetilde{w}^t), \eta_t I_k^t \nabla f_k(w_k^t; \widetilde{w}^t)\rangle + \frac{L}{2}\|\eta_t I_k^t \nabla f_k(w_k^t; \widetilde{w}^t)\|^2\right] \tag{81}$$

$$= \left(\frac{L + \lambda}{2}\eta_t^2 p_k^2 - \eta_t p_k\right)\|\nabla f_k(w_k^t; \widetilde{w}^t)\|^2 \tag{82}$$

$$\leq \left(\frac{L + \lambda}{2}\eta_t^2 p_k^2 - \eta_t p_k\right)2(\mu + \lambda)(f(w_k^t; \widetilde{w}^t) - f(w_k^*; \widetilde{w}^t)) \tag{83}$$

$$\leq \left(\frac{L + \lambda}{2}\eta_t^2 p_k^2 - \eta_t p_k\right)2(\mu + \lambda)(f(w_k^t; \widetilde{w}^t) - f(w_k^*; \widetilde{w}^*)). \tag{84}$$

In the case where $t + 1 \not\equiv 0 \mod E$, i.e. $t + 1$ is not a communication round, $\widetilde{w}^{t+1} = \widetilde{w}^t$. Therefore, we have

$$\mathbb{E}[f_k(w_k^{t+1}; \widetilde{w}^{t+1}) - f_k(w_k^t; \widetilde{w}^t)] \leq \left(\frac{L + \lambda}{2}\eta_t^2 p_k^2 - \eta_t p_k\right)2(\mu + \lambda)(f(w_k^t; \widetilde{w}^t) - f(w_k^*; \widetilde{w}^*)). \tag{85}$$

In the case where $t + 1 \equiv 0 \mod E$, we have

$$\mathbb{E}[f_k(w_k^{t+1}; \widetilde{w}^{t+1}) - f_k(w_k^t; \widetilde{w}^t)] \leq \underbrace{\mathbb{E}[f_k(w_k^{t+1}; \widetilde{w}^{t+1}) - f_k(w_k^{t+1}; \widetilde{w}^t)]}_{\text{B}}$$
$$+ \left((L + \lambda)\eta_t^2 p_k^2 - 2\eta_t p_k\right)(\mu + \lambda)(f(w_k^t; \widetilde{w}^t) - f_k^*). \tag{86}$$

It suffices to bound B:

$$B = \mathbb{E}\left[\frac{\lambda}{2}\|w_k^{t+1} - \widetilde{w}^{t+1}\|^2 - \frac{\lambda}{2}\|w_k^{t+1} - \widetilde{w}^t\|^2\right] \tag{87}$$

$$= \frac{\lambda}{2}\mathbb{E}[\|\widetilde{w}^t - \widetilde{w}^{t+1}\|\|2w_k^{t+1} - \widetilde{w}^t - \widetilde{w}^{t+1}\|] \tag{88}$$

$$\leq \frac{\lambda}{2}\sqrt{\mathbb{E}[\|\widetilde{w}^t - \widetilde{w}^{t+1}\|^2]\mathbb{E}[\|2w_k^{t+1} - \widetilde{w}^t - \widetilde{w}^{t+1}\|^2]} \tag{89}$$

$$= \frac{\lambda}{2}\sqrt{\mathbb{E}[\|\widetilde{w}^t - \widetilde{w}^{t+1}\|^2]}\sqrt{\mathbb{E}[\|(\widetilde{w}^{t+1} - \widetilde{w}^t) + 2(w_k^{t+1} - \widetilde{w}^{t+1})\|^2]} \tag{90}$$

$$\leq \frac{\lambda}{2}\sqrt{\mathbb{E}[\|\widetilde{w}^t - \widetilde{w}^{t+1}\|^2]}\sqrt{\mathbb{E}[\|\widetilde{w}^{t+1} - \widetilde{w}^t\|^2] + 4\|w_k^{t+1} - \widetilde{w}^{t+1}\|^2 + 4\mathbb{E}[\|\widetilde{w}^{t+1} - \widetilde{w}^t\|\|w_k^{t+1} - \widetilde{w}^{t+1}\|]} \tag{91}$$

$$\leq \frac{\lambda}{2}\sqrt{\underbrace{\mathbb{E}[\|\widetilde{w}^t - \widetilde{w}^{t+1}\|^2]}_{C_1}}\sqrt{\mathbb{E}[\|\widetilde{w}^{t+1} - \widetilde{w}^t\|^2] + 4\underbrace{\|w_k^{t+1} - \widetilde{w}^{t+1}\|^2}_{C_2} + 4\sqrt{\mathbb{E}[\|\widetilde{w}^{t+1} - \widetilde{w}^t\|^2]\|w_k^{t+1} - \widetilde{w}^{t+1}\|^2}} \tag{92}$$

where the first and third inequality follows from Cauchy-Schwartz Inequality: $\mathbb{E}[XY] \leq \sqrt{\mathbb{E}[X^2]\mathbb{E}[Y^2]}$. It suffices to find the upper bound of $C_1$ and $C_2$.

$$C_1 = \mathbb{E}\left[\left\|\frac{1}{q}\sum_{k \in S_t} g_k^{t+1}\min\left(1, \frac{\gamma}{\|g_k^{t+1}\|_2}\right) + \beta^t\right\|^2\right] \tag{93}$$

$$\leq \left(\sqrt{\mathbb{E}\left[\left\|\frac{1}{q}\sum_{k=1}^m I_k^{t+1}g_k^{t+1}\min\left(1, \frac{\gamma}{\|g_k^{t+1}\|_2}\right)\right\|^2\right]} + \sqrt{\mathbb{E}[\|\beta^t\|^2]}\right)^2 \tag{94}$$

$$= \left(\sqrt{\mathbb{E}\left[\left\|\frac{1}{q}\sum_{k=1}^m I_k^{t+1}g_k^{t+1}\min\left(1, \frac{\gamma}{\|g_k^{t+1}\|_2}\right)\right\|^2\right]} + \sqrt{d}\sigma\right)^2 \tag{95}$$

$$\leq \left(\sqrt{\frac{m}{q^2}\sum_{k=1}^m \mathbb{E}\left[\left\|I_k^{t+1}g_k^{t+1}\min\left(1, \frac{\gamma}{\|g_k^{t+1}\|_2}\right)\right\|^2\right]} + \sqrt{d}\sigma\right)^2 \tag{96}$$

$$\leq \left(\sqrt{\frac{m}{q^2}\sum_{k=1}^m \mathbb{E}\left[\left\|I_k^{t+1}\eta_t\nabla f_k(w_k^t)\min\left(1, \frac{\gamma}{\eta_t\|\nabla f_k(w_k^t)\|_2}\right)\right\|^2\right]} + \sqrt{d}\sigma\right)^2 \tag{97}$$

$$\leq \left(\sqrt{\frac{1}{m}\sum_{k=1}^m \left\|\eta_t\nabla f_k(w_k^t)\min\left(1, \frac{\gamma}{\eta_t\|\nabla f_k(w_k^t)\|_2}\right)\right\|^2} + \sqrt{d}\sigma\right)^2. \tag{98}$$

Denote $h(t) = \sqrt{\frac{1}{m}\sum_{k=1}^m \left\|\eta_t\nabla f_k(w_k^t)\min\left(1, \frac{\gamma}{\eta_t\|\nabla f_k(w_k^t)\|_2}\right)\right\|^2}$. On the other hand,

$$C_2 \leq \frac{2}{\lambda}\frac{\lambda}{2}\|w_k^{t+1} - \widetilde{w}^{t+1}\|^2 \tag{99}$$

$$\leq \frac{2}{\lambda}f_k(w_k^{t+1}; \widetilde{w}^{t+1}) \tag{100}$$

$$= \frac{2}{\lambda}B_{t+1}. \tag{101}$$

Plug the bounds for $C_1$ and $C_2$ into B:

$$B \leq \frac{\lambda}{2}(h(t) + \sqrt{d}\sigma)\left(h(t) + \sqrt{d}\sigma + 2\sqrt{\frac{2}{\lambda}B_{t+1}}\right) \tag{102}$$

$$\leq \lambda\left(h^2(t) + d\sigma^2 + 2\sqrt{d}\sigma\sqrt{\frac{2}{\lambda}B_{t+1}} + \frac{1}{\lambda}B_{t+1}\right) \tag{103}$$

Denoting the right hand side as $\beta(t)$, we have

$$\mathbb{E}[f_k(w_k^{t+1}; \widetilde{w}^{t+1}) - f_k(w_k^t; \widetilde{w}^t)] \leq \beta(t) + \left((L+\lambda)\eta_t^2 p_k^2 - 2\eta_t p_k\right)(\mu+\lambda)(f(w_k^t; \widetilde{w}^t) - f_k^*). \tag{104}$$

Letting $\delta_k^t = \mathbb{E}[f_k(w_k^t; \widetilde{w}^t) - f_k(w_k^*; \bar{w}^*)]$, we have

$$\delta_k^{t+1} \leq \left(1 - \left((L+\lambda)\eta_t^2 p_k^2 - 2\eta_t p_k\right)(\mu+\lambda)\right)\delta_k^t + \beta(t) \tag{105}$$

Summing over $k$ on the left handed side, when $\gamma$ is large enough so that no clipping happens we have

$$\sum_{k=1}^{m} \delta_k^{t+1} \leq \left(1 - \left((L+\lambda)\eta_t^2 p^2 - 2\eta_t p\right)(\mu+\lambda)\right)\sum_{k=1}^{m}\delta_k^t + m\beta(t) \tag{106}$$

$$= \left(1 - \left((Lp^2 + \lambda p^2 - 2\lambda)\eta_t^2 - 2\eta_t p\right)(\mu+\lambda)\right)\sum_{k=1}^{m}\delta_k^t + m\lambda\left(d\sigma^2 + 2\sqrt{d}\sigma\sqrt{\frac{2}{\lambda}B_{t+1}} + \frac{1}{\lambda}B_{t+1}\right). \tag{107}$$

Let $\Delta_t = \sum_{k=1}^{m}\delta_k^t$. Assume $\max_{t \leq T} B_t = B$. Pick $C = \frac{m\lambda\left(d\sigma^2 + 2\sqrt{d}\sigma\sqrt{\frac{2}{\lambda}B} + \frac{1}{\lambda}B\right)}{\left((Lp^2 + \lambda p^2 - 2\lambda)\eta_t^2 - 2\eta_t p\right)(\mu+\lambda)}$, we have

$$\Delta_{t+1} - C \leq \left(1 - \left((Lp^2 + \lambda p^2 - 2\lambda)\eta_t^2 - 2\eta_t p\right)(\mu+\lambda)\right)(\Delta_t - C). \tag{108}$$

Note that in the case where $t + 1 \not\equiv 0 \mod E$, we have

$$\Delta_{t+1} \leq \left(1 - \left((Lp^2 + \lambda p^2)\eta_t^2 - 2\eta_t p\right)(\mu+\lambda)\right)\Delta_t \tag{109}$$

$$\leq \left(1 - \left((Lp^2 + \lambda p^2 - 2\lambda)\eta_t^2 - 2\eta_t p\right)(\mu+\lambda)\right)\Delta_t. \tag{110}$$

Choose $\eta_t = \eta = \frac{cp}{Lp^2 + \lambda p^2 - 2\lambda}$ for some constant $c$ such that $0 < \left(1 - \left((Lp^2 + \lambda p^2 - 2\lambda)\eta_t^2 - 2\eta_t p\right)(\mu+\lambda)\right) < \frac{1}{2}$. We have

$$\Delta_{t+1} - C \leq \left(1 - \frac{(c^2 - 2c)(\mu+\lambda)}{L + \lambda - \frac{2\lambda}{p^2}}\right)(\Delta_t - C) \tag{111}$$

$$\leq \left(1 - \frac{(c^2 - 2c)(\mu+\lambda)}{L + \lambda - \frac{2\lambda}{p^2}}\right)\left(\left(1 - \frac{(c^2 - 2c)(\mu+\lambda)}{L + \lambda - \frac{2\lambda}{p^2}}\right)^{E-1}\Delta_{t-E+1} - C\right). \tag{112}$$

This is equivalent to

$$\Delta_{t+1} - D \leq \left(1 - \frac{(c^2 - 2c)(\mu+\lambda)}{L + \lambda - \frac{2\lambda}{p^2}}\right)^{E}(\Delta_{t-E+1} - D) \tag{113}$$

where

$$D = \frac{\frac{(c^2-2c)(\mu+\lambda)}{L+\lambda-\frac{2\lambda}{p^2}}}{1-\left(1-\frac{(c^2-2c)(\mu+\lambda)}{L+\lambda-\frac{2\lambda}{p^2}}\right)^E}C \tag{114}$$

$$= \frac{m\lambda\left(d\sigma^2 + 2\sqrt{d}\sigma\sqrt{\frac{2}{\lambda}B} + \frac{1}{\lambda}B\right)}{1-\left(1-\frac{(c^2-2c)(\mu+\lambda)}{L+\lambda-\frac{2\lambda}{p^2}}\right)^E} \tag{115}$$

$$\in \left(m\lambda\left(d\sigma^2 + 2\sqrt{d}\sigma\sqrt{\frac{2}{\lambda}B} + \frac{1}{\lambda}B\right), \frac{m\lambda\left(d\sigma^2 + 2\sqrt{d}\sigma\sqrt{\frac{2}{\lambda}B} + \frac{1}{\lambda}B\right)}{1-\frac{1}{2^E}}\right]. \tag{116}$$

Apply recursively to all $t$, we obtain

$$\Delta_T \le \left(1 - \frac{(c^2-2c)(\mu+\lambda)}{L+\lambda-\frac{2\lambda}{p^2}}\right)^T (\Delta_0 - D) + D \tag{117}$$

$$\le \frac{1}{2^T}\left(\Delta_0 - m\lambda\left(d\sigma^2 + 2\sqrt{d}\sigma\sqrt{\frac{2}{\lambda}B} + \frac{1}{\lambda}B\right)\right) + \frac{m\lambda\left(d\sigma^2 + 2\sqrt{d}\sigma\sqrt{\frac{2}{\lambda}B} + \frac{1}{\lambda}B\right)}{1-\frac{1}{2^E}}. \tag{118}$$

Take $\sigma = \frac{c_2\gamma\sqrt{T\log(1/\delta)}}{m\epsilon}$ and we can find $T = \mathcal{O}\left(\frac{m}{\lambda d\gamma^2}\right)$ such that,

$$\Delta_T \le \frac{1}{2^T}\left(\Delta_0 - \frac{\log(1/\delta)}{\epsilon^2} - \mathcal{O}(mB)\right) + \frac{\frac{\log(1/\delta)}{\epsilon^2} + \mathcal{O}(mB)}{1-\frac{1}{2^E}}. \tag{119}$$

Divide both side by $m$, we have

$$\frac{1}{m}\Delta_T \le \frac{1}{2^T}\left(\frac{1}{m}\Delta_0 - \frac{\log(1/\delta)}{m\epsilon^2} - \mathcal{O}(B)\right) + \frac{\frac{\log(1/\delta)}{m\epsilon^2} + \mathcal{O}(B)}{1-\frac{1}{2^E}}. \tag{120}$$

$\square$

In the non-private case, our Theorem 3 could reduce to the following corollary, which is of independent interest.

**Corollary 8.** *When $\sigma = 0$, Algorithm 1 with $(L+\lambda)$-smooth and $(\mu+\lambda)$-strongly convex $f_k$ satisfies*

$$\frac{1}{m}\Delta_T \le \frac{1}{2^T}\left(\frac{1}{m}\Delta_0 - B\right) + \frac{B}{1-\frac{1}{2^E}}. \tag{121}$$

### A.4 Datasets and Models

We summarize the details of the datasets and models we used in our empirical study in Table 2. Our experiments include both convex (Logistic Regression) and non-convex (CNN) loss objectives on both text (StackOverflow) and image (CelebA and FEMNIST) datasets. We provide anonymized code in the supplementary material for reproducibility. Our code makes use of the FL implementation from the public repo of Laguel et al. (2021) and Li et al. (2021).

Table 2

| Dataset | Number of tasks | Model | Task Type |
|---|---|---|---|
| FEMNIST (Cohen et al., 2017; Caldas et al., 2018) | 205 | 4-layer CNN | 62-class image classification |
| StackOverflow (tff) | 400 | Logistic Regression | 500-class tag prediction |
| CelebA (Liu et al., 2015; Caldas et al., 2018) | 515 | 4-layer CNN | Binary image classification |

## A.5 Hyperparameters

Each fixed privacy parameter $\epsilon$ could be computed by different combinations of noise scale $\sigma$, clipping norm $\gamma$, number of communication rounds $T$, and subsampling rate $p = \frac{q}{m}$. In all our experiments, we subsample 100 different tasks for each round, i.e. $q = 100$, to perform local training as well as involved in global aggregation. For FEMNIST and CelebA, we choose $\sigma \in \{0.02, 0.05, 0.1\}$ and $\gamma \in \{0.2, 0.5, 1\}$. For StackOverflow, we choose $\sigma \in \{0.01, 0.05, 0.1\}$ and $\gamma \in \{0.1, 0.5, 1\}$. We summarize both utility and privacy performance for different hyperparameters below.

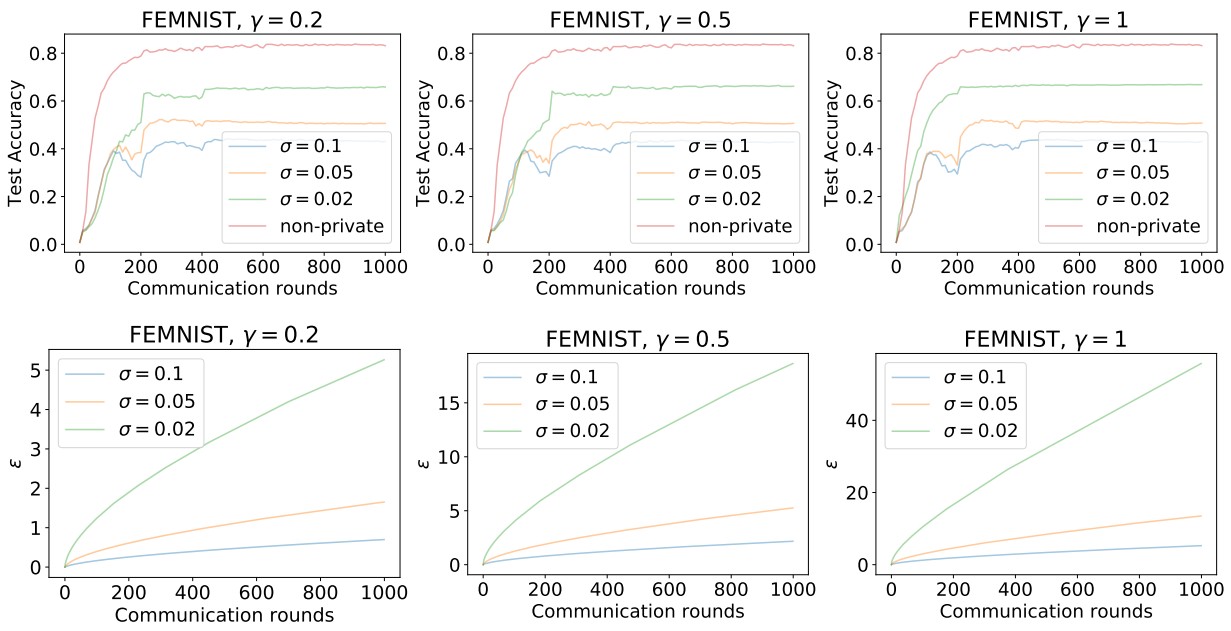

Figure 4: FEMNIST results

## A.6 Comparison with FedProx

Besides FedAvg, we also compared private mean-regularized MTL with other methods that aims to train a global model privately. In particular, we studied private FedProx (Li et al., 2020b) as an alternative global baseline. Note that although the local objective being solved in FedProx is similar to that in mean-regularized MTL, FedProx is a fundamentally different method to handle data heterogeneity in FL from MTL. Specifically, FedProx learns a *global model* where each client solves an inexact minimizer by optimizing local empirical risk with a regularization term. We instead explore learning a *multi-task objective* where each client solves a mean-regularized objective and learns a *separate, client-specific model*. The results are shown in Figure 7. In all three datasets, private FedProx is very similar to private FedAvg under different private parameters $\epsilon$ and worse than private MTL. In particular, in FEMNIST and Stackoverflow, private MTL significantly outperforms training a private global model (FedAvg and FedProx), for all $\epsilon$'s.

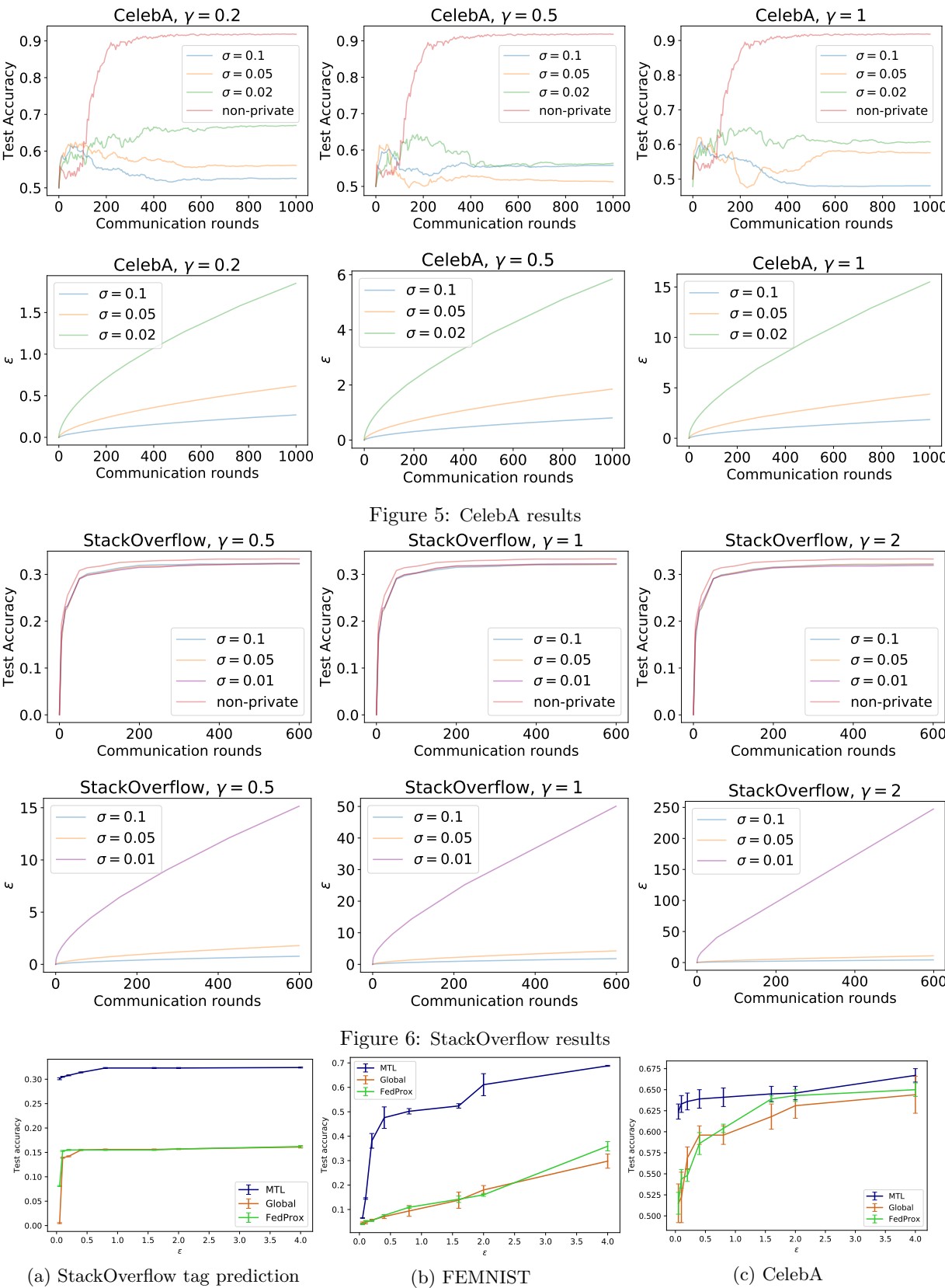

Figure 5: CelebA results

Figure 6: StackOverflow results

(a) StackOverflow tag prediction     (b) FEMNIST     (c) CelebA

Figure 7: Comparison of PMTL and training a private global model(FedAvg/FedProx).

| | Local | PMTL | | | PMTL+best finetuning | | |
|---|---|---|---|---|---|---|---|
| | | $\epsilon = 0.1$ | $\epsilon = 0.8$ | $\epsilon = 2.0$ | $\epsilon = 0.1$ | $\epsilon = 0.8$ | $\epsilon = 2.0$ |
| StackOverflow | .318 | .305 | .323 | .324 | ∗ | ∗ | ∗ |
| FEMNIST | .618 | .371 | .498 | .621 | .663 | .640 | .681 |
| CelebA | .694 | .633 | .641 | .667 | .801 | .817 | .818 |

Table 3: Comparison between PMTL and Local training. For the PMTL+finetuning results on the non-convex problems, we pick the finetuning method that yields the highest test accuracy from all the methods introduced in Section 5.3.

### A.7 Comparison with pure local baseline

While federated learning could yield better utility performance compared to pure local training, this is not always true when we apply client-level DP during federated learning. When a small $\epsilon$ is enforced, accuracy for federated learning could drastically drop (see Section 5). In this section, we compare our PMTL with training pure local model. In addition, since local finetuning does not incur additional privacy cost in our scenario to protect client-level privacy, we also compare PMTL+finetuning with local training. We present the results in Table 3. For StackOverflow where a convex model is used, finetuning with sufficiently many rounds should be the same as training the local model. For the other two datasets where a neural network is trained, training a purely local model performs worse than PMTL under large $\epsilon$ and PMTL with the best local finetuning objective under all $\epsilon$ we evaluated. We note that the goal of our work is not to argue that MTL is better than global/local baselines in all scenarios, but rather to show that it is possible to provide effective private training methods for commonly-used MTL objectives.

### A.8 Comparison to PP-SGD (Bietti et al., 2022)

In this section, we compare PP-SGD (Bietti et al., 2022), a similar form of model personalization in federated learning, with our proposed PMTL on Stackoverflow tag prediction. PP-SGD aims to solve the following *local* objective: $\min_{w,\theta_i} f_i(w, \theta_i, (x, y)) := \ell(y, (w + \theta_i)^\top x)$. It is worth noting that when the model is a neural network, there isn't a straightforward extension for this method to support personalized model weights for each layer, in contrast to our method where the mean-regularization term is calculated by taking the difference of the entire model parameter vector of global and local model. Therefore, for fair comparison, we run PP-SGD on the logstic regression Stackoverflow tag prediction task. Recall that different from the stackoverflow task in the original Bietti et al.

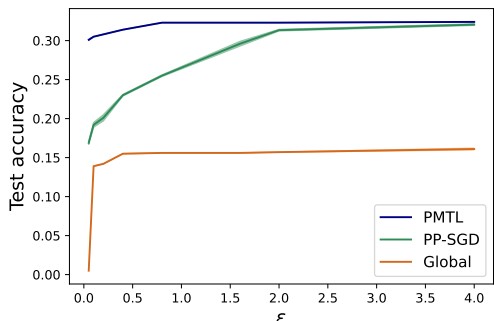

Figure 8: Comparison with PP-SGD on Stackoverflow tag prediction

(2022) paper, we look at a slightly different setting where feature dimension is 10000 and number of classes is 500 (instead of 5000 features dimension and 80 classes in Bietti et al. (2022)). The results are shown in Figure 8. As we see, when we require strong privacy ($\epsilon < 1$), PP-SGD gives a worse privacy-utility trade-off compared to our method. When privacy is weak, our method and PP-SGD achieves similar utility under same privacy budget.

