# OpenReview forum: "Private Multi-Task Learning: Formulation and Applications to Federated Learning"
_TMLR — Accepted by TMLR_

### Review · Reviewer_C5Fx · 2022-12-23

**Summary Of Contributions:**

This paper considers multi-task learning (MTL) under differential privacy/billboard privacy (DP/BP) constraints. They consider the mean-regularized MTL objective and propose using BP to protect client/task privacy. They give a BP algorithm for solving their objective and analyze its convergence. Finally, numerical experiments are provided, showing benefits of their approach vs. training a single global model.

**Audience:**

Yes

**Claims And Evidence:**

No

**Requested Changes:**

Please see **Weaknesses** above.

**Strengths And Weaknesses:**

**Strengths**

-The problem is well-motivated and interesting

-The BP notion seems practical for their MTL problem, although there is some confusion around their definition (see below)

-The paper is well-written for the most part, but there are some aspects that are unclear

-The numerical experiments seem relevant

**Weaknesses**

*Main weaknesses*:
The two main weaknesses that I see are: 1) Def. 2 is not exactly clear, nor is justification for the claim that BP provides strong and meaningful privacy guarantee for MTL (even if other clients collude). 2) Convergence guarantees: the nonconvex one seems to be vacuous (and the strongly convex one is difficult to decipher). I'll start by discussing these points and then list other (more minor) weaknesses.

1) In Def. 2, the notations $f_i$, $g$, and $\Pi_i D$ are never defined as far as I could tell. (Some of these notations become apparent later in the paper.)

This makes the definition unclear and leads me to wonder about the validity of the claim that Def 2 provides strong privacy. For example, if $m=2$, then Def. 2 seems to mean that $\mathcal{M}_1$ is insensitive w.r.t. changes in data set $D_2$? However, if client 2 were an adversarial eavesdropper, then client 2 could use its knowledge of $D_2$ to try to decode $D_1$ since $\mathcal{M}_1$ might depend heavily on $D_1$ (as Def. 2 does not require $\mathcal{M}_1$ to be insensitive to D_1$). Thus, (my understanding of) Def. 2 does not seem to provide a strong guarantee against collusion, as claimed on p.4.

2) In Theorem 3, the assumption that $f_k$ has uniformly $\gamma$-bounded gradient implies that the trivial (DP) algorithm that simply outputs any collection of $w_k$ independent of the data achieves utility of $\gamma^2$ (where utility is measured in terms of average squared gradient norm as in Thm. 3). On the other hand, the second term in Eq. 10 is at least as large as $\gamma^2$ if $B_{i, E} > 1/d$, which would seem to be the case in any interesting/practical application. Thus, it appears that Theorem 3 is vacuous.

*Other weaknesses*

3) last sentence of last bullet on p.2 is a bit unclear: "privacy/utility benefits" relative to what? relative to global (private or non-private?) baseline?

4) p.2: paragraph on MTL background is a bit unclear, as the difference between MTL and federated learning (FL) feels a bit murky here. The authors say that MTL is useful in applications where datasets are shared among  different entities/clients. By contrast, in FL, data is typically decentralized and not shared among clients. Thus, the last two sentences of the paragraph about MTL being useful for cross-device FL by allowing data to be shared was a bit confusing to me.

5) p.2: paragraph about FL: it would be good to cite other works on DP FL here. For example, I suggest citing the following works which are very relevant and seeing the citations therein:

Lowy, Andrew, and Meisam Razaviyayn. "Private federated learning without a trusted server: Optimal algorithms for convex losses." arXiv preprint arXiv:2106.09779 (2021).

Lowy, A., Ghafelebashi, A., & Razaviyayn, M. (2022). Private Non-Convex Federated Learning Without a Trusted Server. arXiv preprint arXiv:2203.06735.

6) brackets in Eq. 2 (inside the sum over $k = 1, \cdots, m$) would clarify

7) Theorem 1: what notion of DP is being claimed here? client-level DP? Be specific.

Also, I was expecting BP here. Maybe lead with Cor. 2 as your main privacy result and mention that you also provide ___ DP..

8) $l_k(w_k)$ in Eq. 8 does not seem to be defined. Earlier, $l_k(x_i, w_k)$ was said to be a loss function but what is $l_k(w_k)$?

9) Theorem 3. Besides being apparently vacuous, it seems that you did not correctly optimize for $T$ in going from Eq. 9 to 10. Also $\mathbb{E}$ is missing on the LHS. Also, you should explicitly state somewhere that since you are assuming uniform $\gamma$-Lipschitz continuity of $f_k$, you do not need to clip in Algorithm 1. Also, it is strange that some terms on the RHS of Eq's 9 and 10 (and LHS of Eq 10) have correct units (scaling with squared gradient norm), while other terms (e.g. the first term on the RHS of Eq. 9) have incorrect units: please either be precise and write each term with the correct scaling factor (preferred) or be informal and omit all scaling factors besides $m, d, \epsilon$.

10) Theorem 5. Similar comments as above with Thm 3. This theorem is strangely written and difficult to parse. I suggest simplifying the bounds to clearly display the dependence on $m, d, \epsilon, \lambda, \mu, B$. Introducing additional notation such as $C$ and $\Delta_T$ is not necessary and makes things less clear. Instead, I expect that you can properly choose parameters, align units, and write the final bound as a sum of 1 or at most 2 terms.

11) Experiments: comparison vs. Bietti et al 2022 would have been nice, as they seem to be the main algorithmic competitor for DP MTL (based on the introduction).

---

> ### Author Response · Authors · 2023-02-15
> **Response to Reviewer C5Fx**
>
> Thank you for your valuable comments and suggestions. We have provided responses to each of the weaknesses (indexed by numbers) suggested by the reviewer.
>
> **1 [Billboard Privacy]**: We have clarified the notations of $f_i,g,\Pi_i D$ in our revision. In the example the reviewer provides, M2 does not see M1. Therefore, in order for client 2 to infer private information about client 1, client 2 can only use their knowledge about D2 and M2. More generally, consider the case where the number of tasks/clients is an arbitrary integer $m$. For every client $i$, even if all the other clients collude, using all information from {Dk} and {Mk} for all k\neq i, the definition protects Di from leakage to all other clients.
>
> **2,9,10 [Convergence guarantees]**: Thanks for your suggestions to improve the presentation of our convergence guarantees. To highlight only the important parameters, we presented an informal version of the convergence result in the main body which we believe may have been the source of confusion. We have revised our Theorem 3 and Theorem 5 to make the upper bound clearer. For the formal, full theorem statement, please refer to Theorem 6 and 7 in the Appendix.
>
> **3 [Privacy-utility benefits]**: The comparison here is relative to the privacy/utility trade-off of learning a single private global model. We have clarified this in our revision.
>
> **4 [MTL vs. FL]**: Thanks for this suggestion; we understand the confusion given that we used the term ‘data sharing’. Here we meant simply that the entire global/population-level dataset is split amongst heterogeneous entities, which also covers the case of FL (where the data is distributed in a heterogeneous manner across a network of devices and each device keeps the raw data local). We have clarified this (in red) in our revision.
>
> **5 [DP FL works]**: Thanks for these suggestions. We have added additional related works (which discuss client-level DP as well as other privacy notions in FL) in our revision.
>
> **6 [Brackets]**: Thank you, we changed this in our revision.
>
> **7 [Theorem 1]**: Yes, we are referring to client-level DP; we have clarified this in our revision.
>
> **8**: $l_k(w_k)$ is short-hand for $\sum_{i=1}^{n_k}l_k(x_i,w_k)$. We have fixed the text to make it clearer.
>
> **11 [Experiments]**: Thanks for the suggestion to compare to Bietti et al., which explores a different form of multi-task learning. We have added a discussion and comparison in Appendix A.8 in our revision. It is worth noting that when the model is a neural network, there isn't a straightforward extension for the method in Bietti et al. (PP-SGD) to support personalized model weights for each layer, in contrast to our method where the mean-regularization term is calculated by taking the difference of the entire model parameter vector of global and local model. Therefore, for fair comparison, we compare to PP-SGD on the (logistic regression) Stackoverflow tag prediction task. Here we find that our approach outperforms PP-SGD, especially in scenarios where we require stronger privacy (e.g., $\epsilon<1$).
>
> Please let us know whether our response has addressed all of your concerns. We are happy to discuss these points further or clarify any additional questions you may have.

---

> > ### Comment · Reviewer_C5Fx · 2023-02-17
> > **Concerns about convergence remain**
> >
> > Thank you for your response. I still have concerns about Theorem 3. Equation 10 in the revision seems to claim a different result (not just superficially) from what was claimed in Equation 10 in the initial submission. For example, the first two terms on the RHS of Equation 10 have different scalings with E, T, and m vs. the respective terms in Equation 10 of the initial submission. This is very strange, especially since I don't see any edits to the formal version of (Theorem 6) or its proof, or the choice of parameters. In its current form, Theorem 6 is very difficult to parse so it is not clear to me whether it is possible to recover (either version of) Theorem 3 from it. Could you please clarify?

---

> > > ### Author Response · Authors · 2023-02-18
> > > **Response**
> > >
> > > Thanks for your quick reply–we apologize for not providing more details about these changes. Equation 10 (Theorem 3) in the original submission and in our revised submission are both simplified/informal versions of the full theorem (Theorem 6, Appendix A.2), which has not changed. In response to your helpful comments, we modified Theorem 3 in our revision to more clearly explain the dependencies on the key parameters and to fix an error that we identified in the informal version of the bound. Here are the details of what was changed:
> > >
> > > - The first term in its complete form is $O(\lambda)*\frac{\sum_{k=1}^m(f_k(w_k^0;\widetilde{w}^0)-f_k^\ast)}{mT}$ (See Eq 29 for details). In our initial submission, we omitted the dependency on $m$ that stems from the summation in the numerator. In our revision we have corrected this, as the term should not be diminishing with $m$: This term should be $\mathcal{O}(\lambda/T)$, as stated in Eq 9. Setting $T=m/(\lambda d\gamma^2)$ thus makes the first term inversely proportional to $m$ (as shown in Eq 10) instead of $m^2$ as in the original submission.
> > > - The second term in its complete form is $O(\lambda)*\frac{\sum_{i=1}^{T/E}B_{iE}}{T}$ (See Eq 29 for details). In our initial submission, we only replaced the $T$ in the denominator and not the numerator (as you can see in our original submission there is still a term of $\sum_{i=1}^{T/E}B_{iE}$). We have made this term simpler in our revision by first setting $B=\max_{i} B_{iE}$, such that $\frac{\sum_{i=1}^{T/E}B_{iE}}{T}$ can be rewritten as $O(B/E)$.
> > > - In the third term we include the dependency of $E$, which is not included in the original version.

---

### Review · Reviewer_iY8Y · 2022-12-28

**Summary Of Contributions:**

This paper studies multi-task learning (MTL) in a distributed environment with privacy guarantees. They show that the standard differential privacy concept does not capture the MTL and propose to use billboard privacy (BP), a variant of differential privacy that fits the scenario. They propose a private multi-task learning algorithm (PMTL) which is shown to satisfy BP and enjoy a standard convergence rate. Empirical evaluation is provided.

**Audience:**

Yes

**Broader Impact Concerns:**

No concerns.

**Claims And Evidence:**

Yes

**Requested Changes:**

My concerns mentioned above are relatively high level and may be difficult to address without a significant change of paper. However, I would like the authors to make the following changes/clarifications:
- Is the algorithm a weaker version of existing implementations of secure aggregation + client DP? E.g. this paper replaces secure aggregation with a trusted global learner.
- Could you explain or convince me the value of introducing BP in addition to existing input privacy + differential privacy?
- A large body of MPC/secure aggregation protocols literature are not discussed in the paper.


**Strengths And Weaknesses:**

### Strengths
This paper brings the billboard privacy for MTL which is an interesting concept to describe the privacy objective.

#### Weakness
My main concern is that the proposed approach is known in the literature while the main technical difficulty is largely ignored in the paper. Please correct me if I am wrong or misunderstand your point.

1. Billboard privacy does not bring much knowledge compared to existing federated learning implementations. The billboard privacy can be seen as a combination of differential privacy and input privacy which has already been implemented in the current federated learning systems (Bonawitz et al., 2019). The input privacy can be achieved through multiparty computation (MPC) protocols or secure aggregation protocols in the FL setting (Bonawitz et al., 2017). The input privacy and output privacy, e.g. differential privacy, are orthogonal concepts and their implementations can be easily combined.

2. This paper assumes there to be a "trusted global learner" which is a very strong assumption. However, this paper only aims to privately optimize objectives like (3). This can be easily done with multiparty computation (MPC) protocols or secure aggregation protocols in federated learning without a "trusted global learner". (And we can add client DP to defend m-1 out of m collusion). The BP concept provides no more insights on how to implement this crucial part which means one can not still circumvent input privacy.

3. The authors claimed that traditional client-level DP does not work, but their proposed algorithm PMTL is essentially traditional client-level DP. In Algorithm 1, one can move the clipping and add noise in line 8 to line 6 and make it client-level DP. It seems to me that the authors are arguing that in order to satisfy Definition 1 you have to choose a very large noise such that the algorithm is useless while in order to reach Definition 2 you can use reasonably small noise. However, definition 1 is certainly not a good privacy concept in the MTL case and is not used in existing implementation ---- again we are using input privacy + differential privacy.

#### Reference
Bonawitz K, Eichner H, Grieskamp W, et al. Towards federated learning at scale: System design[J]. Proceedings of Machine Learning and Systems, 2019, 1: 374-388.
Bonawitz K, Ivanov V, Kreuter B, et al. Practical secure aggregation for privacy-preserving machine learning[C]//proceedings of the 2017 ACM SIGSAC Conference on Computer and Communications Security. 2017: 1175-1191.

---

> ### Author Response · Authors · 2023-02-15
> **Response to Reviewer iY8Y**
>
> Thank you for your valuable comments and suggestions.
>
> **[1. On secure aggregation and the value of BP]**: We agree with the reviewer that input privacy/secure aggregation and client-level differential privacy are orthogonal and could be applied simultaneously. In this work, we focus on the problem of differential privacy as secure aggregation alone does not prevent privacy leakage through the output models. In particular, if all clients in a federated network participate to learn personalized models via multi-task learning, we wish to prevent against any client k inferring the private information of any other client through its trained model. We do this by providing a differentially private formulation for a widely-used family of multi-task learning objectives via billboard privacy, and we study its application to model personalization in federated learning.
>
> Here, billboard privacy is crucial as the notion of traditional client-level DP is incompatible with MTL. In particular, as we discussed in Section 1, Figure 1, and Section 3.2, traditional client-level DP requires that the output of a randomized algorithm be insensitive to changes in the input data. In the MTL setting, this would require that the set of all models be insensitive to changes on any client’s private data. In other words, to protect client-level privacy, traditional DP would force the model for a client’s task to be insensitive to changes in its own dataset---essentially resulting in completely random/uninformed predictions. Although the poor performance of such an approach is intuitive/expected given the definition, we also verify it empirically in Figure 1. Since traditional DP is not an option, we instead propose using BP, a special case of JDP, as a formalization for protecting task-level privacy of an MTL algorithm. As we discuss in Section 3.2, BP is natural to consider as it allows the predictive model for client k to depend on the private data of k while still providing a strong guarantee: Even if all the other m−1 clients collude and share their information, they still will not be able to learn much about client k’s private data.
>
> **[2. Trusted global learner]** The reviewer is correct that we could loosen the assumption of a trusted global learner by considering techniques such as secure aggregation, which is orthogonal to and could be applied in conjunction with our work. We have made note of this in our revision in Section 4. However, we disagree with the reviewer that “we can add client DP to defend m-1 out of m collusion”. The goal of our work is precisely to prevent against m-1 out of m collusion when training multi-task learning objectives, which we show cannot be done with traditional client-level DP while providing reasonable utility and motivates the use of billboard privacy, as discussed above.
>
> **[3. Algorithm 1]** We agree that Definition 1 is not a good privacy concept in MTL. The focus of this work is to provide relaxed privacy definition in the MTL case. However, we want to clarify that if we 'move the clipping and add noise in line 8 to line 6' in Algorithm 1, this would convert global client-level billboard privacy to local client-level billboard privacy, which is still different from traditional client-level differential privacy which requires the output (in this case, *all* m separate models) of a randomized mechanism (SGD training) to be insensitive to changes to *any* single dataset.
>
> **[Discussion on SA]**: Thanks for your suggestion to include a discussion on secure aggregation (SA). We have added a paragraph on MPC/SA in Section 4, highlighted in red. As we have mentioned earlier, we agree that SA/MPC and DP are orthogonal. Our algorithm also has a natural extension to and can be implemented with SA. However, in this paper we focus on providing differential privacy in the context of model personalization (multi-task learning specifically) in federated learning rather than providing new secure computation protocols for FL.
>
> Please let us know whether our response has addressed all of your concerns. We are happy to discuss these points further or clarify any additional questions you may have.

---

### Review · Reviewer_kwvr · 2023-02-01

**Summary Of Contributions:**

In this work, the authors study the multi-task learning problem in a distributed setting under privacy constraints. They (empirically) show that the original definition of differential privacy is not applicable to this problem. So they use a weaker version of differential privacy and developed an algorithm with a convergence guarantee to solve this learning problem.



**Audience:**

Yes

**Claims And Evidence:**

Yes

**Requested Changes:**

Include the Answer to the questions asked in the Strengths And Weaknesses section.

**Strengths And Weaknesses:**

The paper seems well-written, with substantial theoretical study. However, I have the following concerns about the paper,

The authors claim that differential privacy is not applicable to multi-task learning due to reduced performance. If this is true, then is the solution to consider a weaker privacy notion? Since here we are learning m separate models, why not each client train a differentially private model using local data without interacting with others? Do we really need to give up on differential privacy and adopt JDP?

Why does satisfying JDP make sense? To me, it does not make sense. Consider this scenario that each client trains a model independently without differential privacy. In this case, JDP is satisfied.

The convergence bound in (9) does not make sense. As the T goes to infinity, the bound does not go to zero.

Due to the above concerns, the paper needs substantial revision.

---

> ### Author Response · Authors · 2023-02-15
> **Response to Reviewer kwvr**
>
> Thank you for your valuable comments and questions about our work.
>
> **[Why not train m differentially private models using only local data?]**: While it is possible to learn completely separate, client-specific models using only each client’s local dataset, the accuracy of these models can be substantially improved by considering data/models from other clients. This premise is what motivates the problem of federated learning and of multi-task learning more generally. This is also reflected in our experiments: for example, in Table 3, while one could learn local models without incurring client-level privacy costs, we see that the utility is worse compared to that obtained by multi-task learning approaches.
>
> **[Do we need to adopt BP/JDP?]**: The privacy notion we study is client-level (or task-level) differential privacy, whose purpose is to protect every client’s entire local dataset from leakage to other clients. This strong notion of privacy, which is the most common definition considered in *cross-device federated settings* (see, e.g., [1,2]), differs from example-level differential privacy, whose purpose is to protect individual data examples and is typically achieved by applying DP-SGD in machine learning tasks [3].
>
> As we discuss (e.g, Section 1, Figure 1, Section 3.2), traditional client-level DP requires that the output of a randomized algorithm be insensitive to changes in the input data. In the MTL setting, this would require that the set of all models be insensitive to changes on any client’s private data. In other words, to protect client-level privacy, traditional DP would force the model for a client’s task to be insensitive to changes in its own dataset---essentially resulting in completely random/uninformed predictions. Although the poor performance of such an approach is intuitive/expected given the definition, we also verify it empirically (as you note) in Figure 1. Since traditional DP is not an option, we instead propose using BP, a special case of JDP, as a formalization for protecting task-level privacy of an MTL algorithm. As we discuss in Section 3.2, BP is natural to consider as it allows the predictive model for client k to depend on the private data of k while still providing a strong guarantee: Even if all the other m−1 clients collude and share their information, they still will not be able to learn much about client k’s private data.
>
> **[Convergence]**: As we discuss in the last paragraph of Section 4, due to the nature of the mean-regularized MTL objective, the local objective converges to a neighborhood around the optimal. Such behavior is expected and is aligned with prior works that study the same objective but with different solvers (e.g., [4]). We provide an example using simple mean-estimation to illustrate that the convergence lower bound does not diminish with $T$ at the end of Section 4.
>
> Please let us know whether our response has addressed all of your concerns. We are happy to discuss these points further or clarify any additional questions you may have.
>
> [1] Kairouz, P., McMahan, H. B., Avent, B., Bellet, A., Bennis, M., Bhagoji, A. N., ... & Zhao, S. Advances and open problems in federated learning. Foundations and Trends in Machine Learning, 14(1–2), 1-210, 2021.
>
> [2] McMahan, H. B., Ramage, D., Talwar, K., & Zhang, L. Learning differentially private recurrent language models, International Conference on Learning Representations, 2018.
>
> [3] Abadi, M., Chu, A., Goodfellow, I., McMahan, H. B., Mironov, I., Talwar, K., & Zhang, L. Deep learning with differential privacy. In ACM SIGSAC Conference on Computer and Communications Security, 2016.
>
> [4] Hanzely, F., & Richtárik, P. Federated learning of a mixture of global and local models. arXiv preprint arXiv:2002.05516, 2020.

---

### Author Response · Authors · 2023-02-15
**Response to all reviewers**

We would like to thank all the reviewers for their valuable comments and suggestions. We have uploaded a revised version of our draft with revisions highlighted in red. We have also responded to each reviewer separately to address concerns. Please let us know if there is anything that we can further clarify.

---

### Author Response · Authors · 2023-02-28
**Follow up**

Dear Reviewers:

Thanks again for your valuable comments and suggestions. We are just checking to see whether our response have addressed all your concerns. Please let us know if there are further confusions. We are happy to explain and try to address them.

Thank you,
Authors of Submission 660

---

### Decision · Action_Editors · 2023-03-14

**Recommendation:** Accept as is

**Comment:**

In reviewer discussion, all reviewers find the paper somewhat borderline but acceptable as the authors have addressed the concerns raised in the reviews.

**Audience:**

The reviewers believe the paper will be of interest to the community, even though they raise concerns on the practical usefulness of the proposed privacy definition and algorithm.

**Claims And Evidence:**

The paper presents a formulation of privacy suitable for multi-task learning, an algorithm proven to satisfy this definition and a convergence analysis of the algorithm, which appear correct. Simple experiments further support the claimed utility of the method. The reviewers raise no concerns regarding the quality of evidence provided for the claims.